



# TAMSAT-ALERT v1: A new framework for agricultural decision support

Dagmawi Asfaw[1], Emily Black[1], Matt Brown[2], Kathryn Jane Nicklin[3], Frederick Otu-Larbi[4], Ewan Pinnington[1], Andrew Challinor[3], Ross Maidment[1], Tristan Quaife[1]

[1]Department of Meteorology, University of Reading, Reading, UK
[2]Department of Atmospheric, Oceanic and Planetary Physics, University of Oxford, Oxford, UK
[3]School of Earth and Environment, University of Leeds, Leeds, UK
[4]Ghana Meteorological Agency, Accra, Ghana

*Correspondence to*: Dagmawi Asfaw (d.t.asfaw@pgr.reading.ac.uk)

**Abstract.** Early warning of weather related hazards enables farmers, policy makers and aid agencies to mitigate their exposure to risk. We present a new operational framework, Tropical Applications of Meteorology using SATellite data and ground based measurements-AgricuLtural EaRly warning sysTem (TAMSAT-ALERT), which facilitates monitoring of meteorological risk to agriculture. TAMSAT-ALERT combines information on land surface properties, seasonal forecasts and historical weather into quantitative assessments of the likelihood of adverse weather–related outcomes, such as low yield. This article describes the TAMSAT-ALERT framework and demonstrates its application to risk assessment for low maize yield in Northern Ghana. The modular design of TAMSAT-ALERT enables it to accommodate any impact/land surface model driven with meteorological data, and hence to enhance rather than displace existing operational systems. The implementation described here uses the well-established General Large Area Model for annual crops (GLAM) to provide probabilistic assessments of the meteorological hazard to maize yield in northern Ghana throughout the growing season. The results show that climatic risk to yield is poorly constrained at the beginning of the season, but as the season progresses, the uncertainty rapidly reduces. The TAMSAT-ALERT methodology implicitly weights forecast and observational inputs according to their relevance to the metric being assessed. TAMSAT-ALERT can thus be used as a test-bed for the value of probabilistic seasonal forecast information. Here, we show that in northern Ghana, the regional tercile seasonal forecasts of cumulative rainfall and mean temperature, which are routinely issued to farmers, are of limited value for decision making.

## 1 Introduction

Many African people depend on rain–fed agriculture, and are thus vulnerable to drought, and other weather–related hazards. Anticipation of hazard enables farmers and aid agencies to plan ahead, averting disaster (Boyd et al., 2013). Here, we present a new framework for early warning of high meteorological risk to agriculture, the Tropical Applications of Meteorology using SATellite data and ground based measurements-AgricuLtural EaRly warning sysTem (TAMSAT-ALERT). TAMSAT-ALERT integrates assessment of climatological weather-related risk, with forecasts and real-time monitoring of environmental conditions. The framework is intended to be a decision support system, which when combined with socio-economic assessments, can be used by governmental agencies and NGOs to help farmers manage agricultural risk.

The need for timely information on agricultural risk has motivated the development of a number of drought early warning systems and decision support platforms. The Rainwatch-AfClix early warning system (RWX), for example, provides time series of cumulative rainfall, which are compared against historical time years. Users value the facility to compare the current season against past years, finding that it enables them to intuitively gauge risk (Tarhule et al., 2009). The severity of drought, however, depends not only on rainfall. It is, furthermore, not straightforward to translate information on meteorological drought (deficit rainfall) into warning of agricultural drought (deficit soil moisture) (Black et al., 2016). The need to consider



a range of variables, and to compare data from a variety of sources is addressed by more comprehensive platforms, such as the Famine Early Warning Systems Network Early Warning Explorer (FEWSNET-EWX) and International Research Institute (IRI) data library/map rooms, which enable users to compare meteorological data with land-surface remote sensing products, such as Normalized Difference Vegetation Index (NDVI) and soil moisture. Such platforms are aimed at expert users, capable

of interpreting complex, multivariate data. An alternative approach is to use a land surface model, driven with meteorological time series, to derive snapshots and forecasts of soil moisture. The Africa Flood and Drought Monitor (AFDM), for example, estimates soil moisture using a land surface model. The model is driven with satellite data for monitoring current conditions and with bias-corrected, downscaled forecasts, for predicting future conditions (Sheffield et al., 2014). The Africa Flood and Drought Monitor is implemented continent wide, with the aim of monitoring and forecasting metrics related to drought and

flood (soil moisture and streamflow). The AFDM does not, however, attempt to predict crop yield at particular localities. There have been several attempts to forecast yield using crop models, driven by seasonal forecasts (Hansen and Indeje, 2004; Semenov and Doblas-Reyes, 2007). Mismatches between the scales of the input agronomical and climate data, and lack of skill of the seasonal forecasts proved challenging for these early systems (Hansen and Indeje, 2004). In the last few years there have, however, been marked improvements in the skill of sub-seasonal to seasonal forecasts, leading to greater success for

forecasting yield, even in the extra-tropics, where predictability is low. A recent study, for example, demonstrated significant skill for predicting wheat yield in France, using a wheat growing model driven with seasonal forecasts (Canal et al., 2017). Previous operational attempts to predict yield using crop models have mainly focused on issuing predictions in advance of sowing. A weather generator approach to providing continually updated assessments was, however, successfully demonstrated for UK winter wheat yield (Bannayan et al., 2003), indicating the potential of this type of approach for operational risk

assessment.

TAMSAT-ALERT complements existing systems by providing a means of continually updating yield predictions as the season progresses, in a manner similar to that proposed in Hansen et al., 2006 for characterizing the simulated uncertainty in yield, resulting from climatic variability. The TAMSAT-ALERT methodological approach combines the use of historical information, as encapsulated in the RWX methodology, with a land surface/impact model, as demonstrated in the Africa

Drought and Flood Monitor. The system can output any variable or metric that can be generated by the land surface or impact model.

In this study, TAMSAT-ALERT is demonstrated through continually updated seasonal assessments of the meteorological risk to agriculture for Ghana. Although an application of TAMSAT-ALERT has been described elsewhere (Brown et al., 2017), this paper is the first formal description and validation of the methodological approach. Section 2 describes the design of the

framework and give brief notes about its implementation. Section 3 describes the implementation of the framework for assessment of meteorological risk to yield in Ghana. The paper concludes with a discussion of the place that TAMSAT-ALERT has in early warning of meteorological hazard, and wider decision-making processes (Section 4). A user manual for TAMSAT-ALERT is included as supplementary information, and all of the TAMSAT-ALERT scripts are freely available on GitHub.

## 2 Framework concept and design

### 35 2.1 Concept

The TAMSAT-ALERT framework provides a means of deriving quantitative agricultural risk assessments from information on the climatology, historical time series and (optionally) meteorological forecasts. In essence, the system addresses the question:



Given the climatology, the state of the land surface, the evolution of the growing season so far, and (optionally) the meteorological forecast, what is the risk of some adverse event?

The 'adverse event' is any metric that can be derived either directly from meteorological data, or using a model driven with meteorological data. TAMSAT-ALERT is designed to be modular and flexible, enabling users to choose models and datasets
to suit their application. So far it has been applied to risk assessments of agricultural drought using the Joint UK Land Environment Simulator (JULES) model (Brown et al., 2017) and to risk assessments of low yield, using the General Large Area Model for annual crops (GLAM) (section 3.2.2). In addition, code is supplied for assessment of purely meteorological metrics, such as cumulative rainfall (Supplementary information - User Guide).

At a given location and for a given season, the likelihood of an adverse event may depend on past and future weather. Midway
through the growing season, for example, the likelihood of low yield depends both on weather in the past and on the likelihood of unfavourable conditions in the coming weeks. In TAMSAT-ALERT, past weather is based on observations, and future weather is based on the climatology. Thus, a 30-year climatology generates a 30-member ensemble of possible yields, based on 30 possible weather futures, each of which can be driven through a crop model and used to derive a possible yield. Statistical comparison between the forecast ensemble and the climatological ensemble of yield leads to quantitative assessments of the
risk of unfavourable conditions.

In its default set up, for which meteorological forecast information is not included, TAMSAT-ALERT treats all weather futures as equally likely. The risk assessments can, however, be refined by weighting the ensemble members, based on probabilistic forecast information - for example, tercile forecasts of cumulative rainfall or mean temperature, cumulated/averaged over a 90-day period. Specifically, the value of the metric being forecasted for each ensemble is used to assign that ensemble member
to a particular tercile. Each ensemble member is then weighted by the appropriate tercile probability (see section 2.2 for further explanation). If there is a weak link between the metric being forecast (for example regional seasonal rainfall) and the risk being assessed (for example local low yield), then the forecast will have little impact on the risk assessments. Conversely, if the link is strong, skilful forecasts can significantly reduce the uncertainty in the risk assessments. TAMSAT-ALERT is thus both a method for downscaling/bias correcting meteorological input into impact models, and a method for accounting for
mismatch between forecast variables and metrics of risk.

There are several sources of potential predictive power in TAMSAT-ALERT. Firstly, as the season progresses, the amount of observational information included in the forecast increases, and the range of possible outcomes is thus reduced. Secondly, the antecedent state of the land surface (especially root zone soil moisture) has a significant effect on the likelihood of drought, and hence low yield (Brown et al., 2017). Thirdly, local information on the climatology determines the likelihood that
meteorological conditions will be sufficiently favourable during the remainder of the season to offset less favourable past meteorological and land-surface conditions. Finally, skilful meteorological forecasts provide direct information on the likelihood of adverse weather conditions in the remainder of the season. The relative importance of these sources depends on the metric being predicted, along with the local climate and land-surface conditions. The effect of meteorological forecasts depends both on the precision/skill of the forecast and the relevance of the meteorological forecast metric for the metric of
hazard assessed by TAMSAT-ALERT.





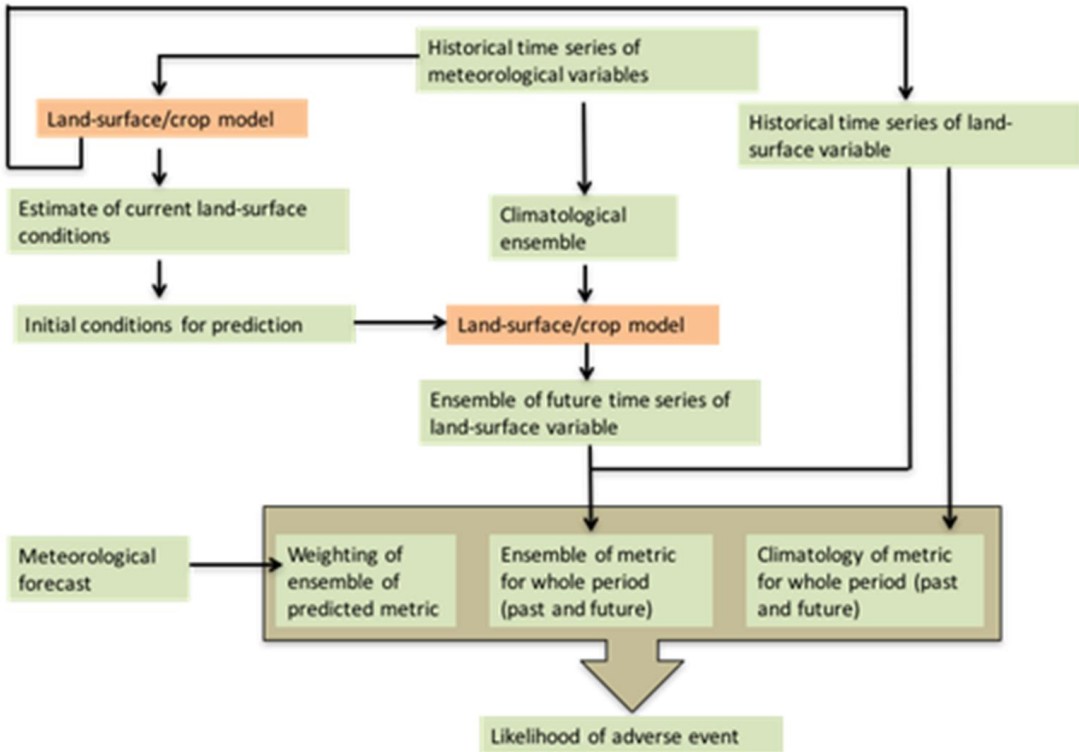

**Figure 1: Conceptual overview of the TAMSAT–ALERT system.**

## 2.2     Model implementation

The TAMSAT-ALERT framework is illustrated in Figure 1.  The user provides a time series of driving data, which is long

enough to generate a statistically meaningful ensemble and climatology.  The driving meteorological data is used in several ways: to generate an ensemble of predictions; to assess the progress of the period of interest so far and to derive initial conditions for the future period (if required for the ensemble predictions); and to generate a climatology against which the forecast ensemble can be compared.  Once the climatology and ensemble have been produced, meteorological forecast information is optionally introduced, to weight the ensemble members.  The system is modular and thus easily adapted for

different impact models, metrics of risk and meteorological forecasts.

The steps for deriving probabilistic assessments of risk of some adverse event on a particular day (the day in question) can be summarized as follows:

1. The user prepares a file containing historical time series of driving data, along with any other parameter files (e.g. agronomical or soil parameters). These should extend at least until the day in question. Note that TAMSAT-ALERT

v1.0, only supports daily input. Support for higher/lower resolution data will be introduced in future versions of the framework.

2. The user converts the long daily time series of driving data into the appropriate format for their impact model, and carries out a historical run, in order to derive an annual historical time series of their chosen risk metric. This enables a base line assessment of climatological risk. The risk metric time series should be presented as an annual time series

of the form <year> <data>. Here, we will call this time series file 'historical_metric.txt'.





3.  For the probabilistic risk assessments, the impact model is driven with an ensemble of meteorological forcing data, generated by TAMSAT-ALERT. As described earlier, the period of interest might contain both the past and the future.

    a.  For the past, the meteorological driving data for ensemble member includes identical time series, taken from observations.

    b.  For the future, the meteorological driving data for each ensemble member is based on the historical climatology. Specifically, for a given Day of Year (DoY), the driving data are taken for that DoY for a year in the past. To maintain the daily weather statistics and the consistency between variables, each ensemble member is based on a particular past year. Thus, ensemble member 'x' is based entirely on year 'y'.

    To accomplish this, the TAMSAT-ALERT system converts the daily time series of driving data into multiple files, each containing driving data for one ensemble member. The user is allowed to set the period over which the ensemble system will be run. This is distinct from the period over which the metric is calculated (the metric period). The period over which the ensemble will be run should include sufficient time before the metric period to allow for spin up. The user makes any format changes necessary to convert these TAMSAT-ALERT driving data files into driving data specific to their impact model. The user then carries out the ensemble prediction runs, driving their impact model with the TAMSAT-ALERT processed time series of driving data. The user impact model should be set up to output annual time series of their metric of interest. This ensures that each ensemble member is associated with the year for which the possible weather future was derived (see above). The output can thus be presented in a single file, with two columns: <year> <data>. Here, we will call this file ensemble_metric.txt

4.  The risk assessment is derived by comparing the mean and standard deviation of the climatological baseline distribution (historical_metric.txt derived in (2)) with the mean and standard deviation ensemble distribution (ensemble_metric.txt derived in (3)). Note that an alternative option, employing an empirical cumulative distribution function is also provided by the TAMSAT-ALERT system. The ECDF approach is suitable for non-Gaussian variables, but can result in noisy predictions if the ensemble is relatively small.

    At this point, meteorological forecast data is incorporated (if available):

    a.  An annual historical time series of the metric being forecast (e.g. cumulative June-August rainfall) is provided by the user. Here, we will call this weighting_metric.txt, which is of the form <year>, <data>. The data series should be provided for the years used to generate the weather future aspect of the ensemble (i.e. ensemble_metric.txt as described in (3b)). [The TAMSAT-ALERT v1.0 release includes a utility function for extracting forecast metrics from the historical driving meteorological data file supplied by the user.]

    b.  The annual time series of forecast metric is then ranked. Based on this ranking, each historical year is assigned to a forecast category. In the case of terciles, in the previous example, the bottom third of JJA rainfall years is assigned to tercile 1, the middle third to tercile 2 and the top third to tercile 3.

    c.  As was noted in 3b, each ensemble member is associated with a historical year and ensemble_metric.txt is presented in the form <year> <data>. Each data point in this file can thus be associated with a quantile category, using the year assignments described in 4b.

    d.  When calculating the mean and standard deviation, the ensemble is weighted by the user-supplied categorical forecast probabilities, which are assigned to each member during 4c.



The TAMSAT-ALERT code is written in Python. All code and documentation (including a user manual) for TAMSAT-ALERT have been released on GitHub (https://github.com/tamsat-alert/v1-0). However, users need to have their own working installations of their chosen impact model. The TAMSAT-ALERT v1.0 release consists of scripts to:

- Convert meteorological time series into driving data for both the ensemble forecast.
- Calculate quintile predictions for user defined risk metrics, based on the input files historical_metric.txt, ensemble_metric.txt and weighting_metric.txt.
- Produce a set of plots comparing the ensemble and climatological distribution (see User Guide in supplementary information).

In the GitHub release, in addition to the general TAMSAT-ALERT framework scripts listed above, scripts are provided that set TAMSAT-ALERT up for (i) for the GLAM crop model (the implementation demonstrated in Section 3 of this paper), and (ii) for assessments based purely on time means/accumulations of meteorological variables. A test case is provided so that users can be assured that the system is working as expected.

## 3  Demonstration of the system: A case study of maize yield prediction in Ghana

This case study demonstrates the use of TAMSAT-ALERT system for forecasting the risk of poor maize harvest in Ghana.
The first and second part of the case study describe the study area and the implementation and evaluation of a mechanistic crop model GLAM. The third part demonstrates the implementation of GLAM as part of the TAMSAT-ALERT system for continually updated risk assessments.

### 3.1  Study area

Ghana is located on the southern coast of West Africa, between latitudes 4° 44' N and 11° 11' N and longitudes 3° 11' W and
1° 11' E. Rainfed agricultural systems are the major component of the Ghanaian economy, accounting for 30 % of the GDP and employing half of the labour force (PARI, 2015). The country is divided into six agro-ecological zones, each with a distinct rainfall pattern (Figure 2). The Northern part is dominated by Guinea Savannah with average annual rainfall of 1000–1100 mm from one rainy season spanning May to September, while in the southern part, moist semi-deciduous agro ecology dominates, with an average annual rainfall of 1500 mm, falling within two rainfall seasons (Owusu and Waylen, 2009; Owusu
and Waylen, 2013). Most of the cereal crops (primarily Sorghum, Millet and Maize) are produced in the northern part of Ghana (Martey et al., 2014). Table 1 shows the six agro ecological zones with the average annual rainfall and major crops grown in the agro-ecological zones.

Table 1: Characteristics of agroecological zones in Ghana (source: www.fao.org/nr/water/aquastat/countries_regions/GHA/)

| Agro ecological zone | Rainfall (mm annum$^{-1}$) | Number of seasons | Major crops grown |
|---|---|---|---|
| Sudan savannah | 1000 | 1 | Millet, Sorghum, Maize |
| Guinea savannah | 1100 | 1 | Maize, Sorghum |
| Transition zone | 1300 | 1 | Maize, Roots, Plantain |
| Moist semi deciduous forest | 1500 | 2 | Roots, Plantain |
| Costal savannah | 800 | 2 | Roots, Maize |
| Rainforest | 2200 | 2 | Roots, Plantain |

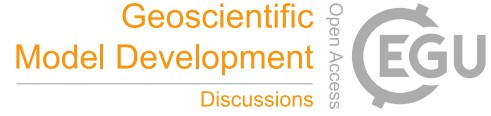



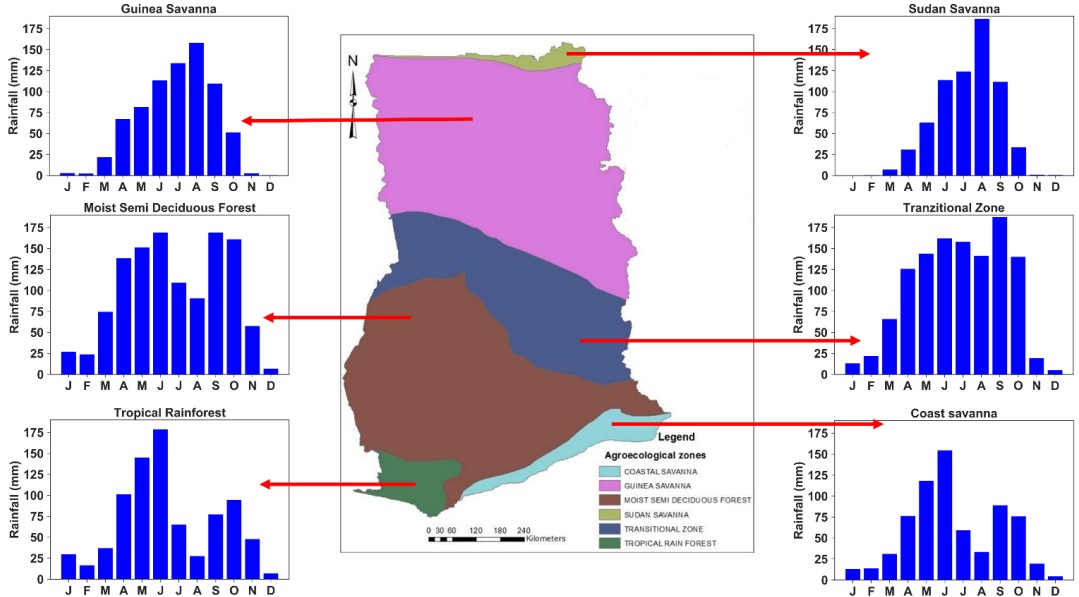

**Figure 2: Agro Ecological zones of Ghana (source: Sidibe et al., 2016) and average seasonal rainfall pattern of each agro ecological zone, based on TAMSAT rainfall estimates.**

Maize is one of the major crops produced in Ghana. The production area and the amount of yield has been increasing since

5  1994 (Figure 3). Figure 4 shows a time series of maize yield in Ghana (expressed in Kg ha$^{-1}$). From 1994–2006 there is no observed trend but after 2007, there is a step change in yield, coinciding with the introduction of a new variety by the Crop Research Institute (CRI) of the Council for Scientific and Industrial Research (CSIR) of Ghana in 2007 (Ragasa et al., 2013).

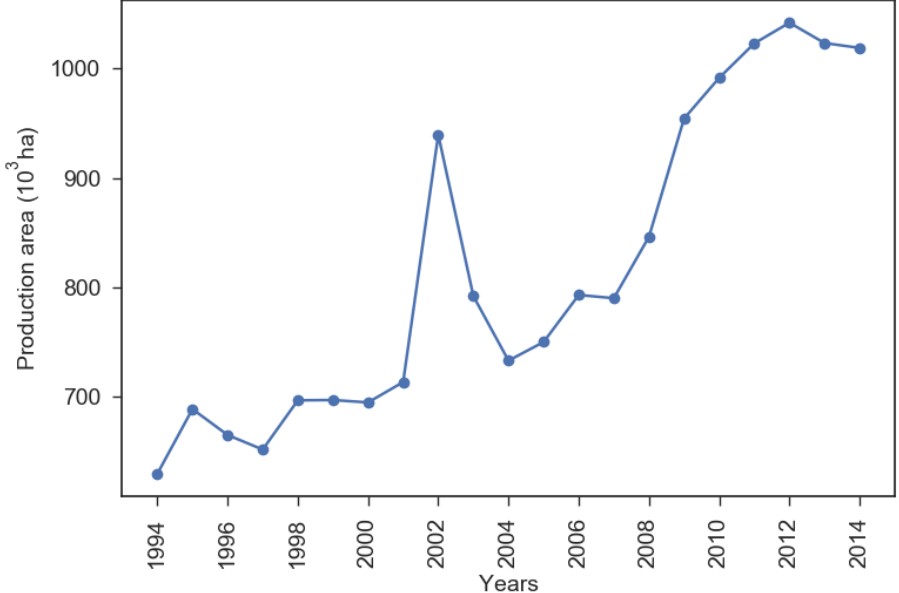

**Figure 3: Maize production area over Ghana from 1994 to 2014.**





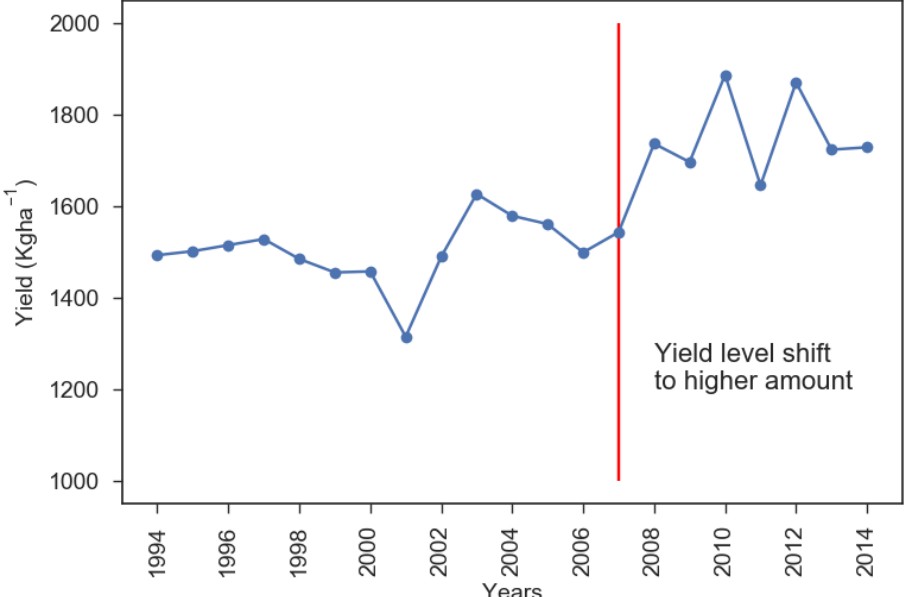

**Figure 4: Maize yield in Ghana 1994–2014. There are two separate periods marked by the red lines where we observe changes in yield. 1994–2006 there is no clear trend in the yield produced and 2007–2014 there is a shift in the production where higher yield is observed.**

**3.2    Data and methods**

**3.2.1    Datasets used**

The driving weather data sets for the evaluation of the model were daily time series extracted from the Watch Forcing Data ERA-Interim (WFDEI) (Weedon et al., 2014) for shortwave radiation, maximum temperature, minimum temperature and rainfall. For the demonstration of the system at a point, the driving data were based on daily, quality controlled station data

provided by the Ghana Meteorological Agency. The station used is Tamale which is located Northern Ghana (9.41° N, 0.85° W). Precipitation and maximum/minimum temperature were measured directly, and shortwave radiation was derived from sunshine hours.

Tercile forecast data was downloaded from the publicly available IRI regional forecasts (http://iri.columbia.edu/our-expertise/climate/forecasts/seasonal-climate-forecasts/). The IRI forecasts are based on a hybrid dynamical/statistical method,

developed by the U.S National Oceanographic and Atmospheric Administration North American Multi-Model Ensemble Project (NOAA-NMME) (Kirtman et al., 2014). The seasonal forecasts are issued at the beginning of each month for precipitation and temperature at a global scale with a spatial resolution of 2.5° for precipitation and 2° for temperature (Barnston and Tippett, 2014). The IRI forecasts were chosen for this analysis because of their wide use by African meteorological services and regional climate outlook forums.  In this study the seasonal forecast data were used in the form

they are supplied to farmers - i.e. tercile probabilities of 3-month cumulative rainfall / 3-month mean temperature at a regional level.

In addition to meteorological time series, GLAM requires data on soil type and the agronomical properties of maize (Section 3.2.2). For this study, the soil texture was set to be sandy loam and planting date was set starting from 124th day of the year to 154th day of the year which allows a 30-day planting window. The maize agronomical properties were taken from the published

literature, and are presented in the supplementary information (Table S1).





GLAM was evaluated against national level maize yield data released by the FAOSTAT (http://www.fao.org/faostat/) (see Figure 4). Although the FAO issues guidance on the compilation of these datasets, in practice, there is little quality control and the data should be treated with caution.

### 3.2.2    The GLAM crop model

As described in section 2.1, the TAMSAT-ALERT system can be used to assess any metric of risk that can be output by a model driven with meteorological data. In this study, the General Large Area Model (GLAM) for annual crops is used to simulate maize yield and subsequently to monitor the probabilistic risk of poor harvest as the growing season progresses.

GLAM is a process based crop simulation model, which incorporates sufficient processes to capture the impact of climate variability on crop yield (Challinor et al., 2004; Ramirez-Villegas et al., 2015b). GLAM uses a limited number of driving data
sets and an intermediate complexity of crop development process representation. Nevertheless, previous studies have demonstrated that GLAM has skill in capturing the impact of weather on crops (Challinor et al., 2005; Challinor et al., 2006). Such information enables users to translate time series of weather into a time series of yield estimates (Challinor and Wheeler, 2008). GLAM has also been used to model weather and climate change impact on crop yield and adaptation strategies (Parkes et al., 2015; Ramirez-Villegas et al., 2015a; Ramirez-Villegas and Challinor, 2016).

GLAM requires daily values of precipitation, shortwave radiation, maximum temperature and minimum temperature as driving weather data with additional inputs of soil properties and planting window (Watson et al., 2013). GLAM accumulates the above ground biomass, which is a product of daily transpiration and predetermined transpiration efficiency value, within the growing season to determine total biomass production which is converted into yield using a harvest index (Osborne et al., 2007). Planting date is either prescribed by the user, or determined using GLAM's intelligent planting date system (the
approach taken in this study). It is important to note that GLAM does not account, in a process-based fashion, for non-meteorological influences on crop growth, such as pests, diseases and fertilizer use. Rather, these factors are encapsulated in the yield gap parameter (YGP), which is determined by calibrating the model yield with observed yield (Challinor et al., 2004). The YGP is assigned a value between zero and one where 'one' represents the potential yield, given the weather conditions, soil texture and crop development parameters (Challinor et al., 2005).

### 3.2.3    GLAM model evaluation

GLAM was used to simulate the yield from 1994 to 2014 using the Watch Forcing Data Era-Interim (WFDEI) as a driving dataset. The WFDEI has a 0.5º by 0.5º resolution and so GLAM was output at this resolution. The simulated yield at each grid point was then weighted by the year 2000 season fraction of production area over each grid point to make a country average yield (Weedon et al., 2014; Monfreda et al., 2008). This country average yield was then compared with the FAO maize yield
data set for the same period. It was shown on Figure 3 that maize production can be split into two distinct periods: 1994–2006 and 2007–2014. Because of the reported changes in agronomic practice and drought tolerant maize variety introduction through the Drought tolerant maize for Africa (DTMA) project in 2007, (Obeng-Antwi et al., 2013; Ragasa et al., 2013) the transpiration efficiency (TE) value was increased from 7.0 for the period 1997–2006 to 8.0 for the period 2007–2014. The YGP was maintained at 0.4 for the whole simulation period.


The results of the simulated crop yield are presented on Figure 5 and the statistical values of the comparison are presented with the scatter plot on Figure 6. GLAM was able to maintain the overall mean yield and, and as a result, the normalized root mean square error (NRMSE) is very low (0.07). The overall correlation value is found to be 0.67 (Pearson) and the Spearman correlation which is less affected by outliers is 0.8. The difference in the Spearman and Pearson correlation coefficients is
mainly due to the severe overestimation of 2001 season yield, probably resulting from a long dry spell, the impact of which



on farming practices was not fully accounted for by GLAM (FAO/WFP-GIEWS, 2002). Some of the correlation strength is due to capturing the change in mean yield from 1994–2006 to 2007–2014 period, and this is done by changing transpiration efficiency (TE) value for the two periods. The strength of the correlation in the two individual periods specified above suggests that the link between Ghana–wide weather and yield is moderate – an important consideration for policy makers when they

5    make use of information from TAMSAT-ALERT. This is primarily due to the myriad of factors that can affect yield, including agronomical practice, pests and disease and socio-economic problems. Nevertheless, in vulnerable regions, the meteorological risk to yield is, in itself, an important consideration for agricultural agencies because action can be taken to mitigate the hazard. This might include subsidizing drought resistant varieties or encouraging early planting/re-planting.

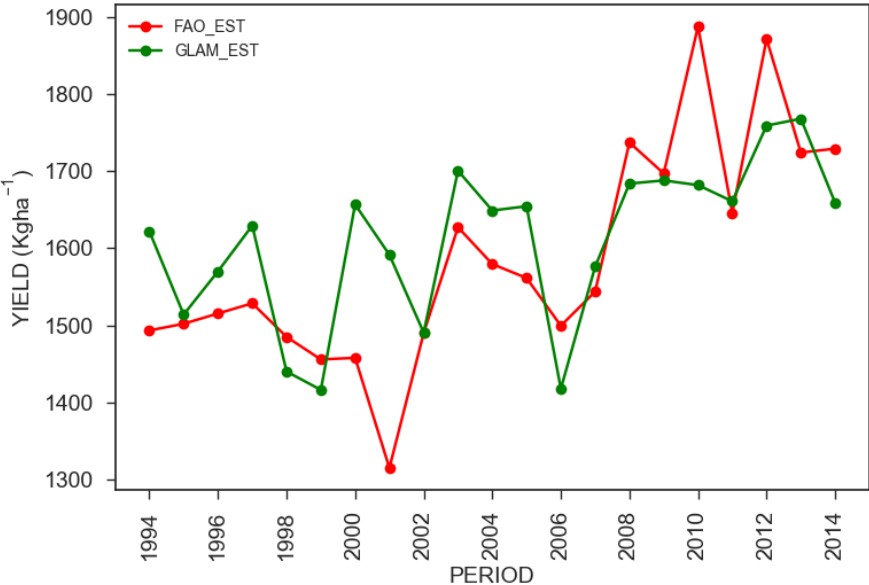

10   **Figure 5: Time series of FAO yield (red line) and GLAM simulated yield (green line).**

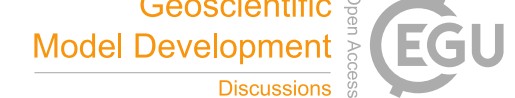

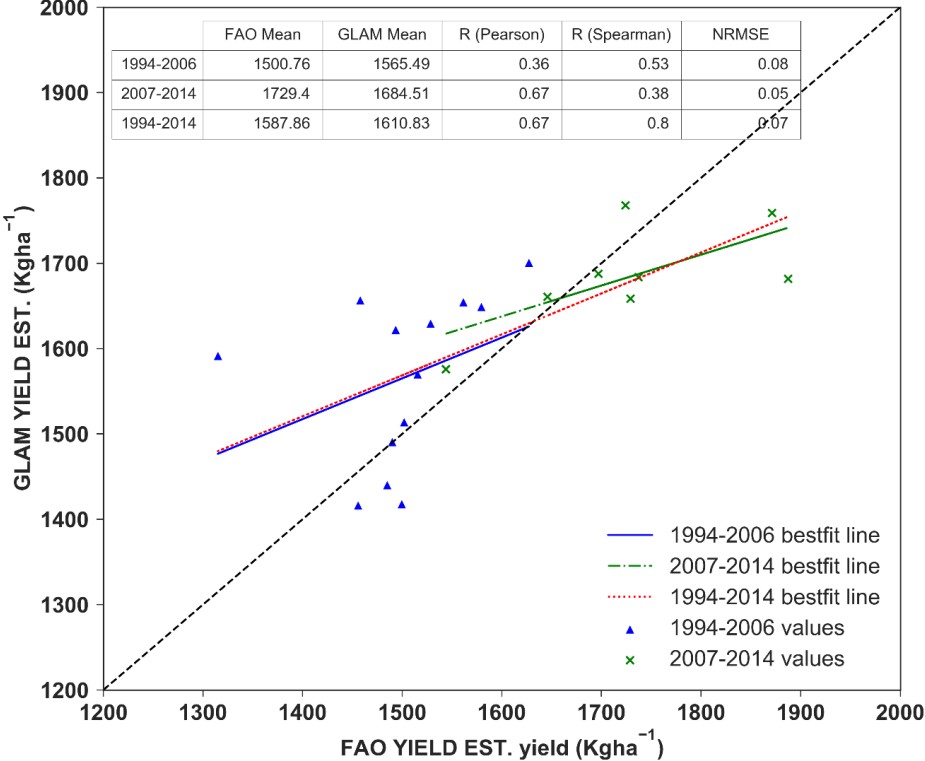

**Figure 6: Scatter plot between FAO yield and GLAM simulated yield. The red dotted line is the best fit line for the whole period considered (1994–2014). The blue solid line shows the best fit line for the period 1994–2006. The green line shows the best fit line for the period 2007–2014.**

### 3.2.4    Incorporation of GLAM into TAMSAT–ALERT

Figure 7 shows how GLAM has been incorporated into the TAMSAT-ALERT system. As described in Section 2, time series of driving data based on historical observations are used both to derive climatological yield and to generate an ensemble of predicted yield. Individual planting dates are determined for each ensemble member using GLAM's intelligent planting date system, and the crop is harvested when the growing degree days requirement is fulfilled (Challinor et al., 2004; Challinor and Wheeler, 2008). Because of the way TAMSAT-ALERT is set up to incorporate observational data continually as the season progresses, once the optimum planting time is passed for the year being hindcast, the planting date for each ensemble member converges. Analogously, once the harvest date for the hindcast year has passed in the observations, the harvest date, and indeed the predicted yield, for each ensemble member is identical. In this implementation of GLAM, a climatological period of 30 years (1980–2009) was used for the yield forecast.



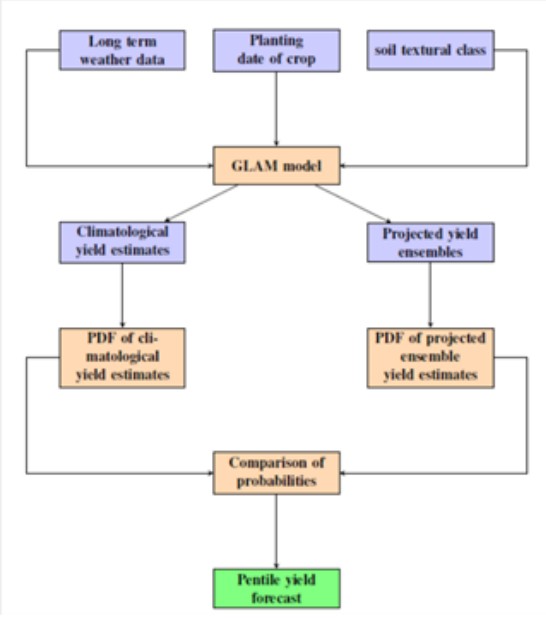

**Figure 7: Process flow chart for crop yield forecasting within the TAMSAT-ALERT system. The blue boxes represent input data sources, the orange colour represent the processes involved in the system and the green box show the final probabilistic forecast for the crop yield.**

### 3.3    Case study results

### 3.3.1    Yield forecasting using GLAM: 2011 season example for a single location (Tamale)

The following case study describes the implementation of TAMSAT-ALERT at a single location for which we have high quality meteorological station data (Tamale). The use of a single location enables us to illustrate TAMSAT-ALERT's role in decision making at the community level.

Figure 8, Figure 9 and Figure 10 illustrate the implementation of TAMSAT-ALERT for the 2011 growing season – nationally, a low yield year compared to other post-2007 years (noting that we do not have yield data for Tamale). The hindcasts were initiated every 5 days. GLAM infers that planting occurred on 4th June and harvesting on 15th September to 20th September. Figure 8 depicts all ensemble members in the context of the climatological spread in yield.  Figure 9 shows histograms of ensemble members at monthly intervals, starting 10 days after planting. Figure 10 shows a time series of ensemble spread
(standard deviation of ensemble yield predictions).

At the outset of the season, the yield estimates are derived only from the meteorological climatology; no in-season observational data is incorporated. The spread is thus large (equivalent to the climatology). During the season, as in-season data is incorporated by TAMSAT-ALERT, the meteorological time series driving GLAM become progressively more similar. As a result, the ensemble rapidly converges. In this example, for instance, two months after planting, the ensemble standard
deviation is 34 % of the climatology.



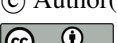

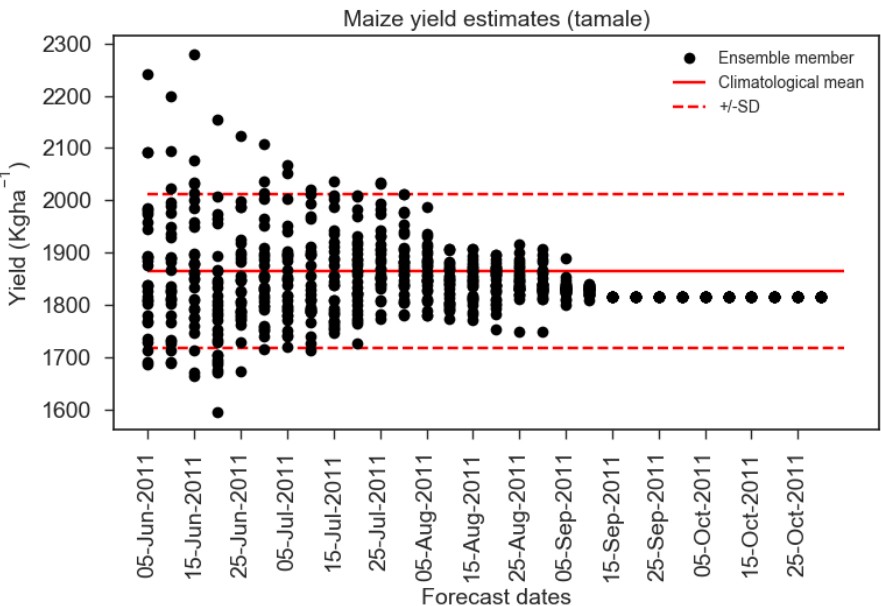

**Figure 8: An example of hindcast of maize yield using GLAM implemented into the TAMSAT-ALERT system. Black dots represent individual ensemble members and red lines are the climatology.**

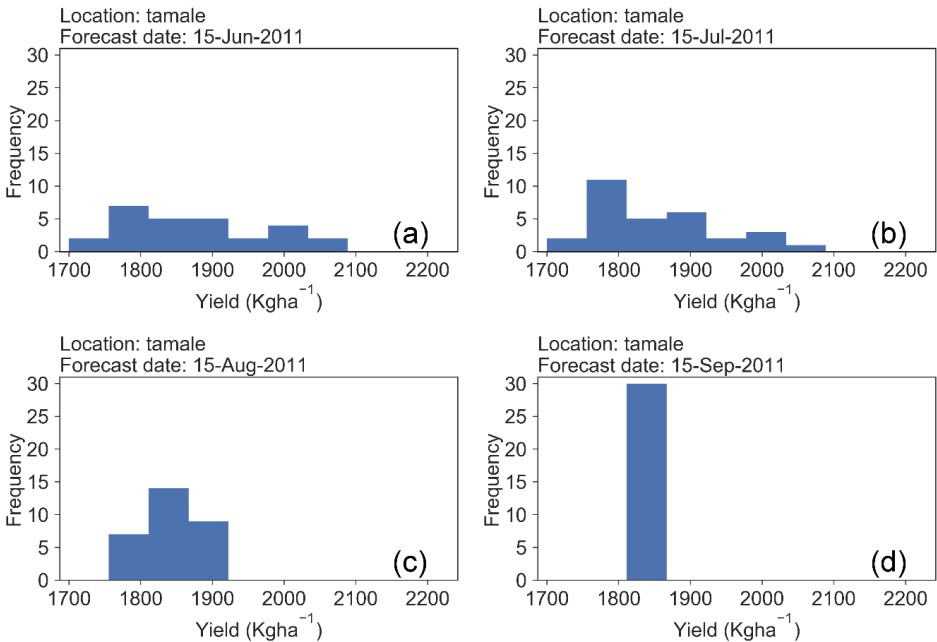

5    **Figure 9: Histograms of yield forecast for (a) 15 June 2011, (b) 15 July 2011, (c) 15 August 2011, and (d) 15 September 2011.**



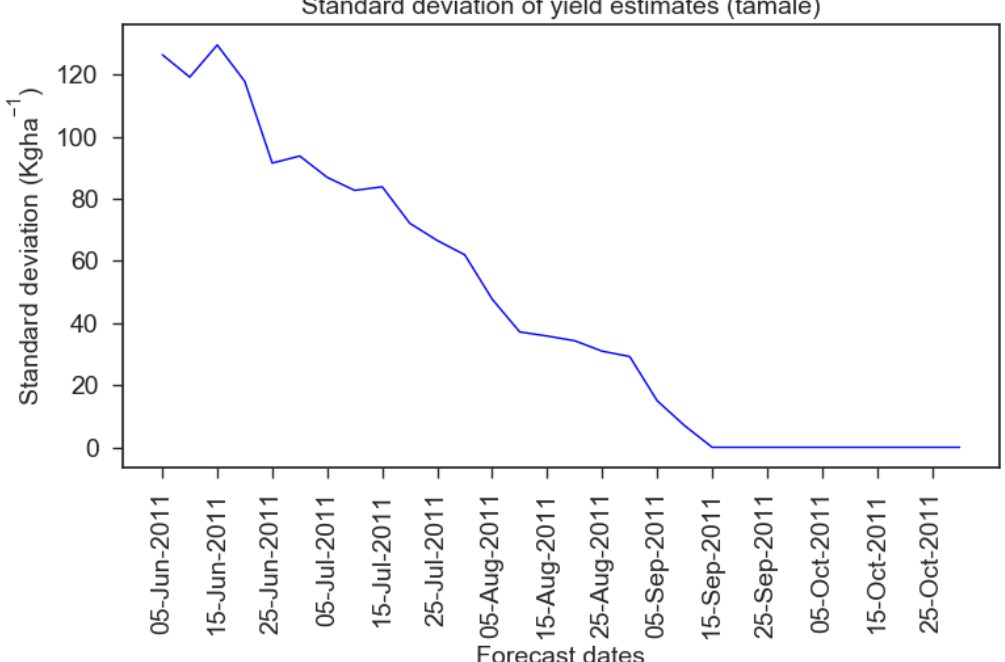

**Figure 10: Standard deviation of the yield estimate initiated on the dates displayed on the x-axis.**

The yield forecasts can be communicated with end users in a probabilistic form, with the ensemble expressed as quintiles, representing the following categories: above the 80th percentile, 60th–80th percentile, 40th–60th percentile, 20th–40th percentile

and below 20th percentile. These categories can be equated to very high, high, average, low, and very low yield respectively. An example of such quintile forecasts at monthly intervals during the 2011 growing season is shown on Figure 11. Consistent with Figures 8 and 9, at the outset of the season, the categories are equally likely except the extreme categories, the difference in probability coming from the change in planting date for some years in the climatological period considered (1980–2009). As the season progresses, the average and low categories become more likely and the extreme categories (very high and very

low), less likely.

It is evident from Figure 11 that the ensemble mean tends towards average or low values, even 2 months ahead of the harvest date during 2011 – suggesting a degree of precision, even towards the beginning of the growing season. Section 3.3.3 presents a formal evaluation of skill for the 2002–2011 period.

### 3.3.2    Incorporation of meteorological forecasts

As described in Section 2.1, the TAMSAT-ALERT framework can use probabilistic information from meteorological forecasts to weight the yield forecast ensemble – providing a means of incorporating forecast information into the decision support system. In this study, we consider tercile forecasts of cumulative 90-day rainfall and mean 90-day temperature to reflect the information currently available to the Ghana Meteorological Agency. To illustrate the process of including forecasts, we continue with the 2011 case study. We have used idealized perfect tercile seasonal forecasts for total June-July-August (JJA)

precipitation to weight the forecast on 4 June 2011, July-August-September (JAS) precipitation to weight the forecast on 4 July 2011, August-September-October (ASO) precipitation to weight the forecast on 4 August 2011, and September-October-November (SON) precipitation to weight the forecast on 4 September 2011.





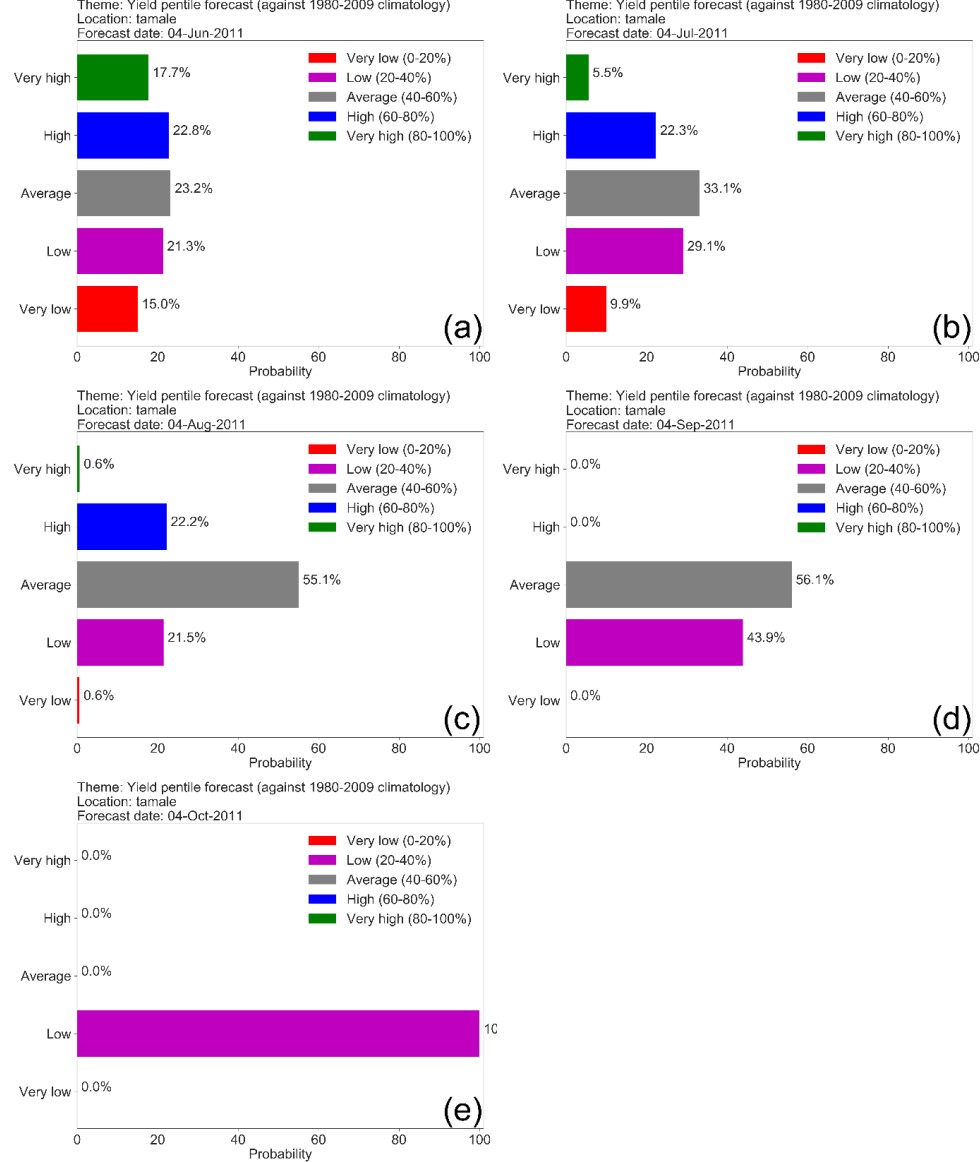

**Figure 11: Probabilistic forecasts for maize yield in Northern Ghana (Tamale) for five dates (a) 4 June 2011, (b) 4 July 2011, (c) 4 August 2011 and (d) 4 September 2011 and (e) 4 October 2011. The planting date was 4 June 2011. In the first day of planting the impact of the weather is not indicated well that the yield probabilities are spread more or less equally in all categories but after one month in 4 July 2011 it is indicated that 62 % of the ensembles fall in the average and low categories after two months 4 August 2011 76 % of the ensembles indicate an average and low yield estimate compare to the climatological yield. Few days before harvest on 5 September 2011 100 % of the yield is estimated to be in the average and low quintile category.**

To assess the potential value of tercile rainfall and temperature seasonal forecast information, we have weighted the ensemble as if the next 90 days of temperature and cumulative rain are known (i.e. perfect forecast experiment). So, we consider three probabilistic forecasts: tercile weightings of [0,0,1] for the lower, middle and upper tercile respectively (perfect wet forecast), [0,1,0] for the lower, middle and upper tercile respectively (perfect normal forecast) and [1,0,0] for the lower, middle and upper tercile respectively (perfect dry forecast). The ensemble was weighted by these perfect tercile forecasts according to the actual total rainfall (perfect rainfall forecast) or the actual mean temperature (perfect temperature forecast) that ensued in the next 90 days following each TAMSAT-ALERT hindcast.




Figure 12 shows the yield forecast probabilities when the perfect rainfall forecast is used. When a perfect rainfall forecast is used to weight the ensemble, the probabilities of the quintile forecast show more rapid convergence especially two months into the season. The improvement is less noticeable for the first forecast date and one month into the season because of the low correlation between metric being forecast (cumulative rainfall over the 90 days following the risk assessment date), and

5  the maize yield.

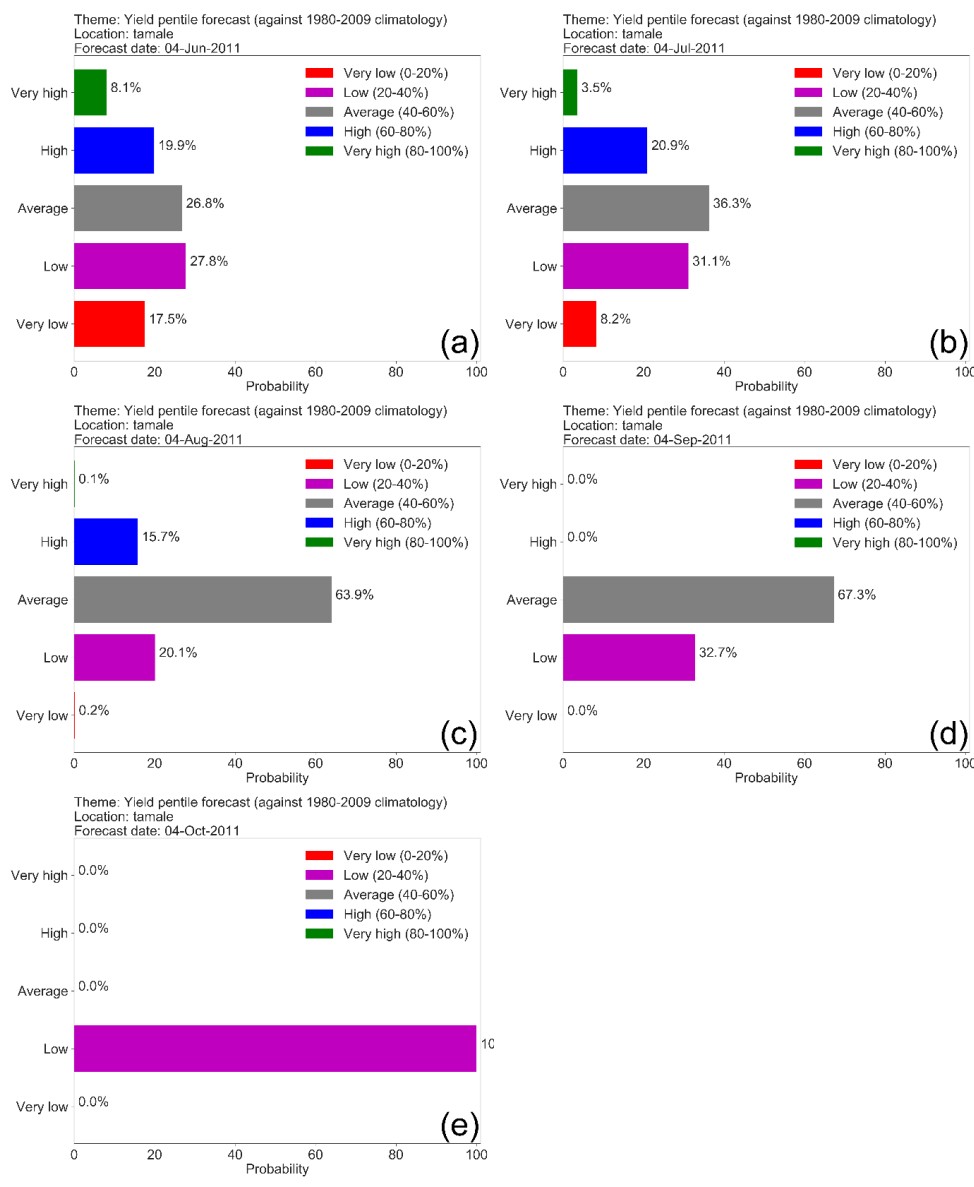

**Figure 12: Yield probability forecast for the year 2011 for five forecast dates (a) 4 June, (b)4 July, (c) 4 August, (d) 4 September and (e) 4 October when ensembles are weighted by a perfect tercile seasonal rainfall forecast.**

An alternative approach is to use temperature forecasts to weight the ensemble. To investigate the effect of temperature

10  forecasts, the ensemble was weighted using idealized, perfect June-July-August (JJA) tercile temperature forecasts to weight the risk assessments on 4 June 2011, July-August-September (JAS) tercile temperature forecasts to weight the risk assessments



on 4 July 2011, August-September-October (ASO) tercile temperature forecasts to weight the risk assessments on 4 August 2011, and September-October-November (SON) tercile temperature forecasts to weight the forecast on 4 September 2011. As with rainfall, the upper, middle and lower terciles are weighted [1,0,0] for a 'perfect cold forecast', [0,1,0] for a 'perfect normal forecast' and [0,0,1] for a 'perfect warm forecast'. Figure 13 shows the forecast for the 2011 cropping season with a perfect

5    average temperature forecasts. Due to a negative correlation of the average temperature with maize yield, a warmer temperature forecast is associated with predictions of lower yield. Comparison between Figures 12 and 13 suggest that temperature forecasts have greater effect on the risk assessments than rainfall forecasts.

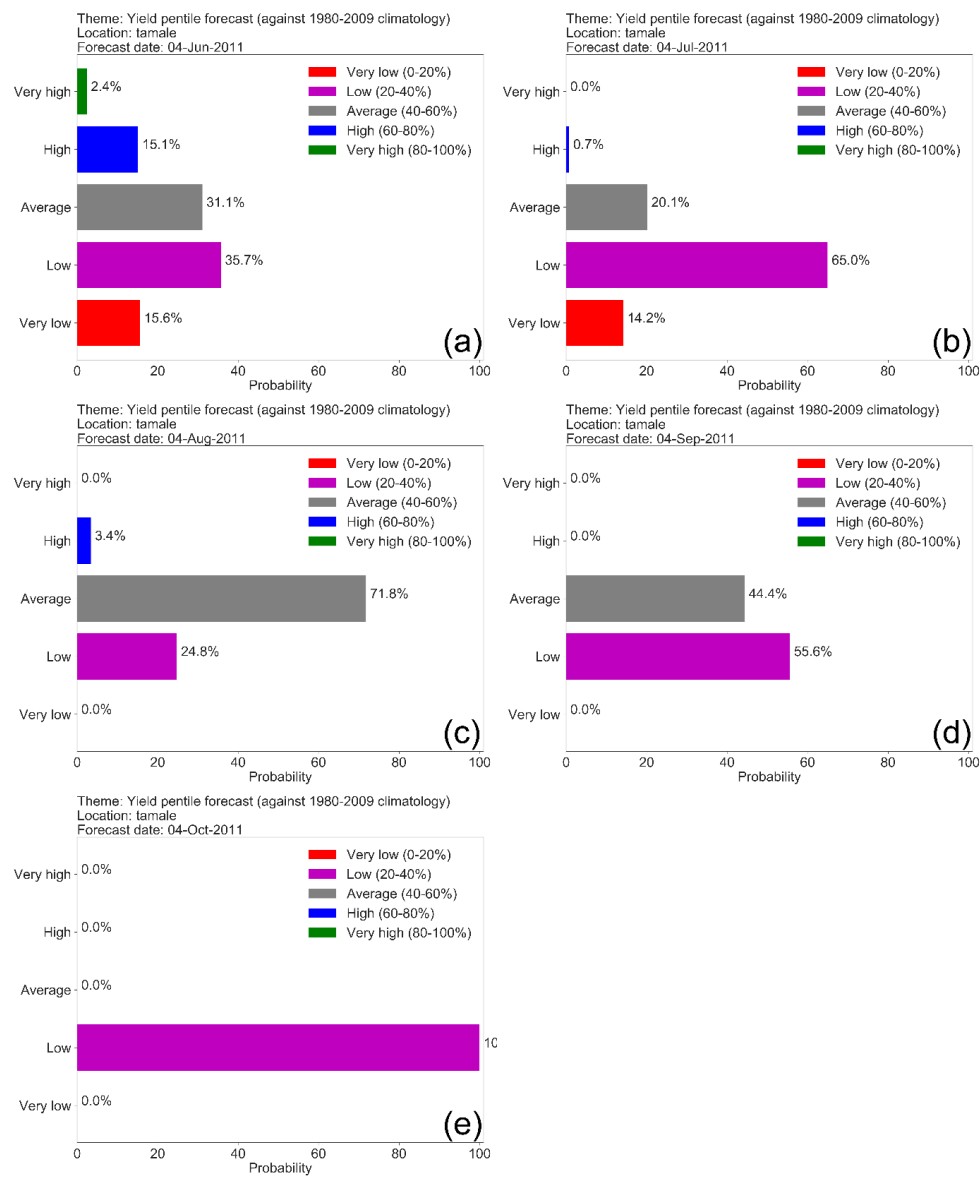

**Figure 13: Yield probability forecast for the year 2011 for five forecast dates (a) 4 June, (b) 4 July, (c) 4 August, (d) 4 September**
10   **and (e) 4 October when ensembles are weighted by a perfect average temperature of seasonal forecast.**

So far, only idealized forecasts have been considered. In the next section, we demonstrate the effect of using actual tercile forecast information issued by the International Research Institute (IRI) for rainfall and temperature. The seasonal forecast from IRI for 2011 in northern Ghana are shown on Table 2.





**Table 2: IRI tercile seasonal forecast for the 2011 season.**

| Season | Rainfall | | | Temperature | | |
|---|---|---|---|---|---|---|
| | Below normal | Normal | Above normal | Below normal | Normal | Above normal |
| JJA | 25 | 35 | 40 | 30 | 40 | 30 |
| JAS | 33.3 | 33.4 | 33.3 | 30 | 40 | 30 |
| ASO | 33.3 | 33.4 | 33.3 | 45 | 35 | 20 |
| SON | 33.3 | 33.4 | 33.3 | 33.3 | 33.4 | 33.3 |

Figure 14 shows the yield forecast probabilities based on weighting the yield ensembles by seasonal rainfall forecasts from

5    IRI. Comparison with Figure 11 suggests that the weighting has little effect. Figure 15 shows the quintile yield predictions when temperature forecast weightings from IRI are applied. As with rainfall, comparison with Figure 15 shows that the weighting has little effect.

The results are summarized on Figure 16, which represents the probability of each yield pentile at different lead time of the 2011 season yield forecast, with no seasonal forecast, precipitation forecast, and temperature forecast applied. For all lead time

10   periods indicated weighting by IRI seasonal forecast for the 2011 season showed no improvement in predicting the final yield compared to the non-weighted values. This is not surprising. The tercile weightings for the IRI forecast (Table 2) are close to climatology, and the previous discussion, moreover, showed that even a perfect and precise seasonal forecast of 90-day cumulative rainfall/mean temperature has relatively little impact.



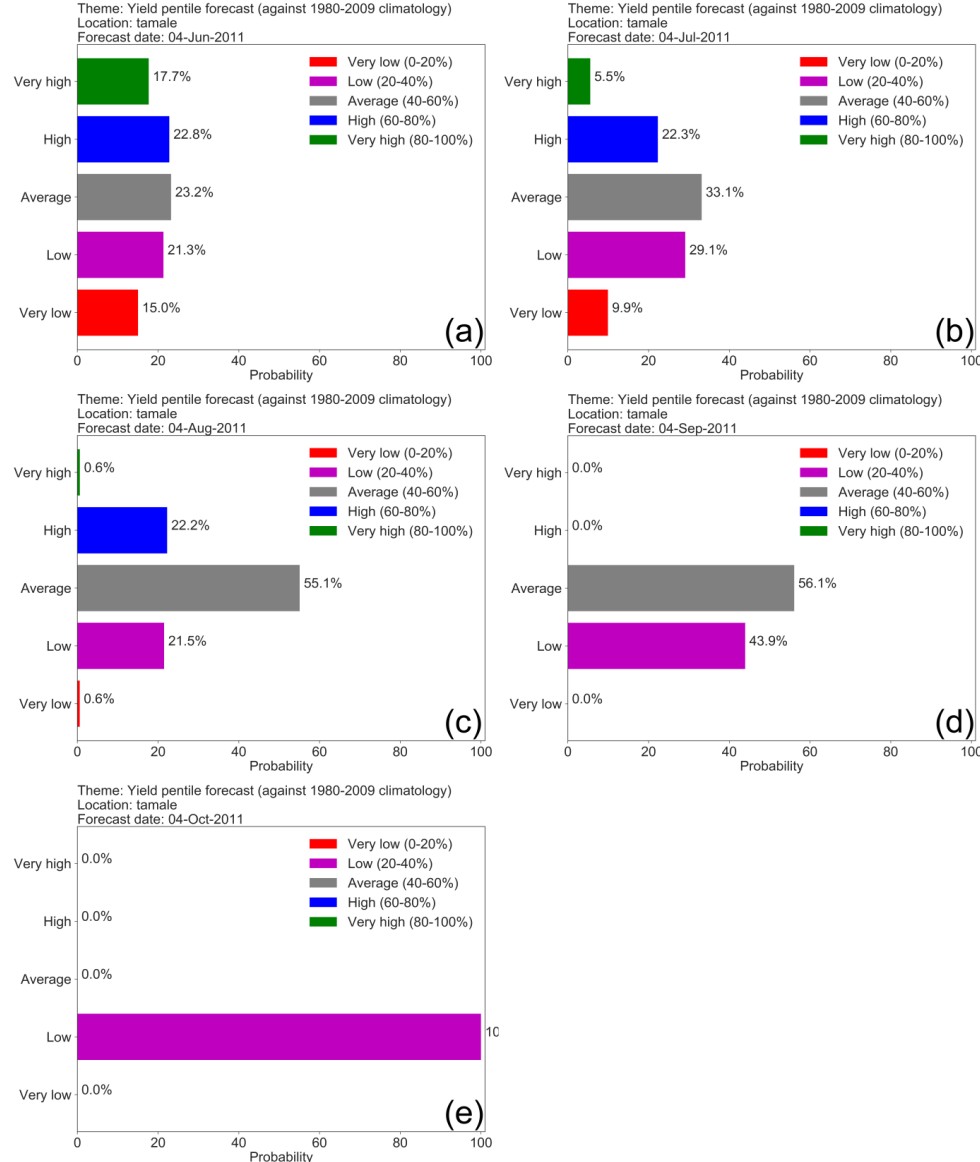

**Figure 14: Yield probability forecast for the year 2011 for five forecast dates (a) 4 June, (b) 4 July, (c) 4 August, (d) 4 September and (e) 4 October when ensembles are weighted by IRI seasonal rainfall forecast.**





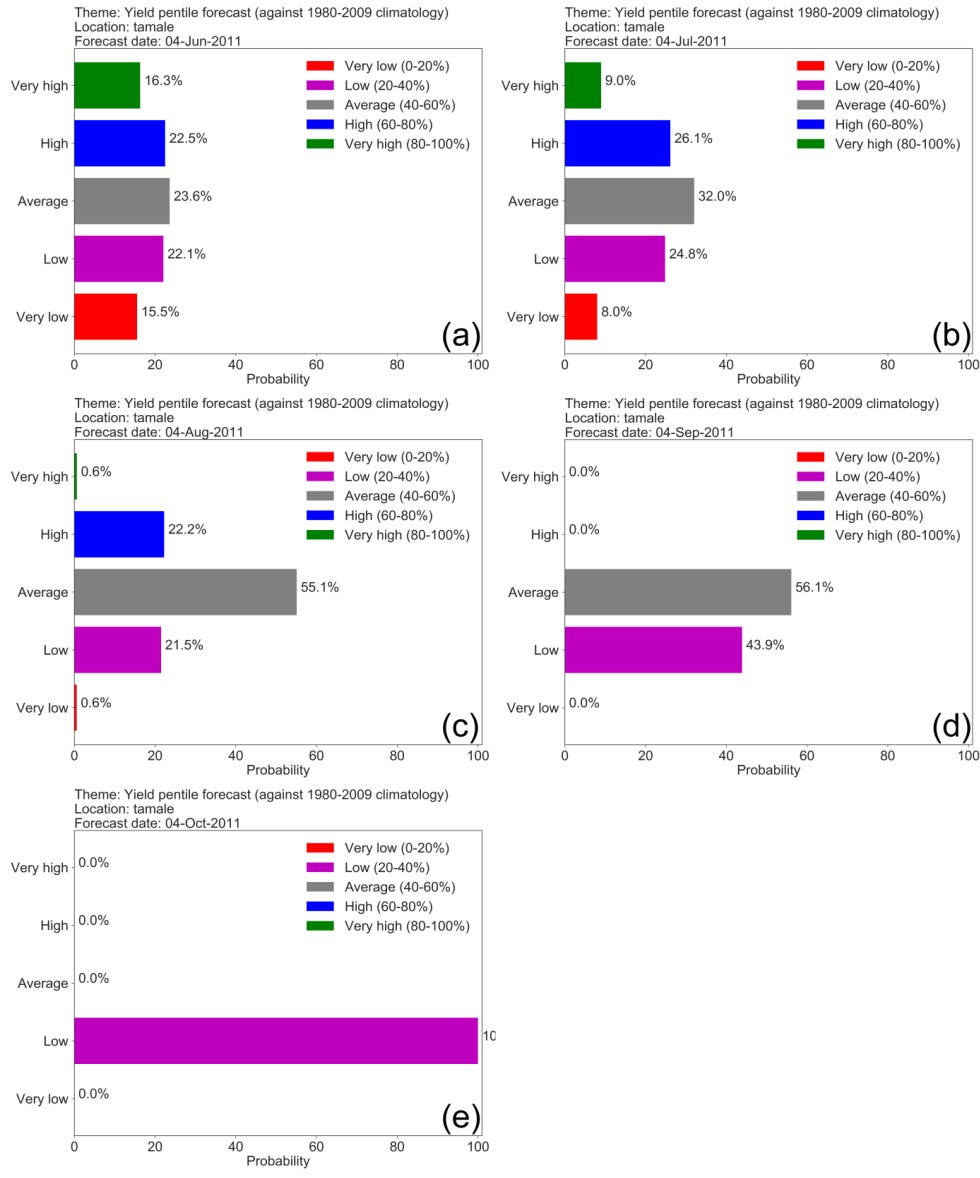

**Figure 15: Yield probability forecast for the year 2011 for five forecast dates (a) 4 June, (b) 4 July, (c) 4 August, (d) 4 September and (e) 4 October when ensembles are weighted by IRI seasonal forecast average temperature.**



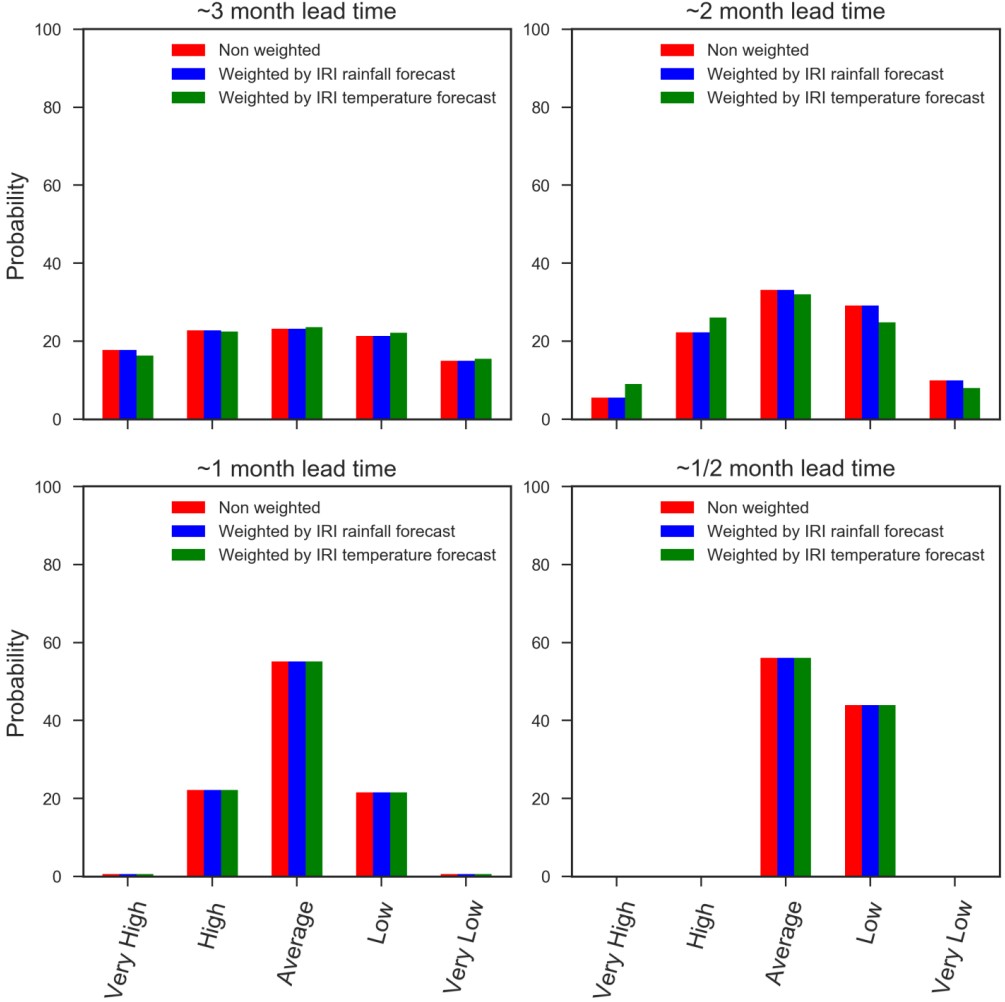

**Figure 16: Probability of yield forecast for 2011 growing season when weighted by IRI seasonal forecast of rainfall (blue), when weighted by IRI seasonal forecast of temperature (green) and no weightings are used (red). The x-axis represents the pentile categories used in the yield forecast.**

5 **3.3.3     Formal skill evaluation**

The objective of TAMSAT-ALERT is to provide early warning of the meteorological risk to yield, which is not an observable quantity. For this reason, the evaluations of TAMSAT-ALERT's skill are carried out in a 'perfect model' framework, in which we attempt to forecast the yield simulated by GLAM forced with observed weather data. It is important not to confuse these skill assessments with evaluation of GLAM (Section 3.2.3) – although the usefulness of the framework depends, to a large

10    extent, on the quality of the model and data incorporated within it.

Figure 17 shows GLAM hindcasts at four approximate lead times (i.e. ~3, ~2, ~1, ~1/2 months ahead of harvest), for five years. Towards the outset of the season, the hindcasts for each year are similar, and close to the climatology, with the minor differences explained by variation in planting date. For all the lead times considered the spread of the ensembles reduces as the season progresses.



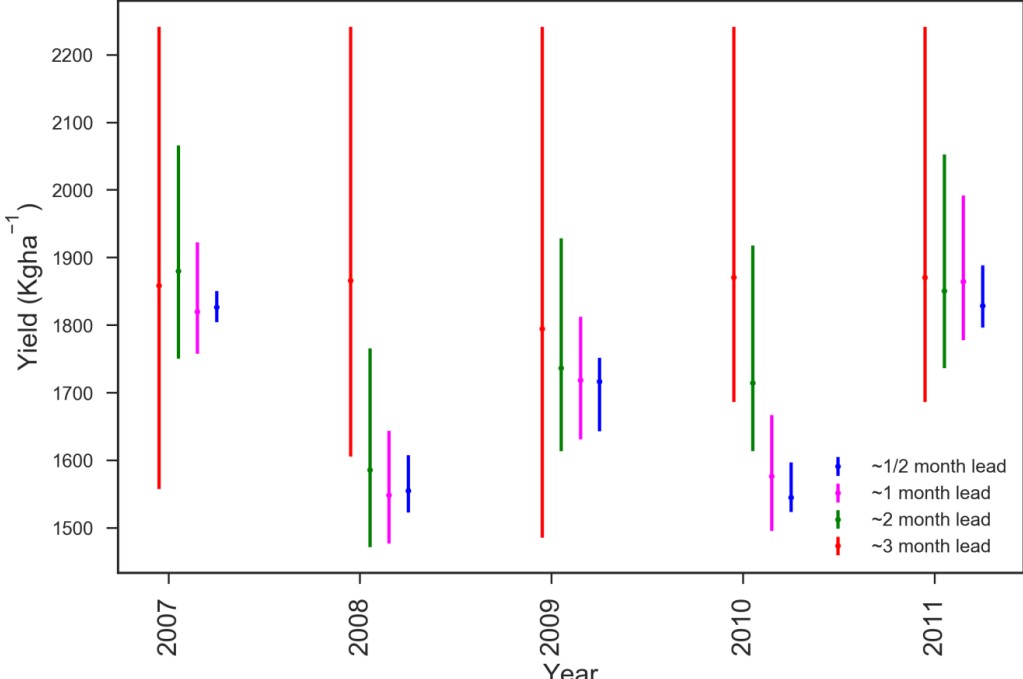

**Figure 17: Time series of maize yield forecast in Ghana from 2007 to 2011 with four lead time of forecast. This is done using a hindcast for each year and comparing the plots of ~3–month lead time (red), ~2–month lead time (green), ~1–month lead time (magenta) and ~1/2–month lead time (blue).**

As described in Section 3.3.1, the probabilistic ensemble forecasts will be presented as the likelihood of quintile categories. The skill of the probabilistic forecast was assessed using the ranked probability skill score (RPSS). The ranked probability skill score is a skill score formulated from the ranked probability score (RPS) that compares the cumulative squared probability error for forecasts and climatological forecast for each category identified. The RPSS is negatively biased with smaller ensemble sizes ($< 40$) and due to this a correction was done on the reference RPS used before calculating the final RPSS. The

bias corrected RPSS is called the discrete ranked probability skill score ($RPSS_D$). Details of the calculation and bias correction are given in (Muller et al., 2005; Weigel et al., 2007). Positive values indicate better skill than the climatology; a unit value represent a perfect score and zero or below zero values indicate no skill in the forecast.

The $RPSS_D$ for Tamale was derived for the period 2002–2011. This period is used because IRI seasonal forecasts for

precipitation and temperature issued on monthly basis are only available from 2002. Figure 18 indicates the skill scores for the four lead times for the forecasts made using the TAMSAT-ALERT system. The skill scores are generally above 0.4 for ~2-month lead time and over 0.6 for ~1-month lead time over the 10-year period considered. There are some years where the skill score was lower than the stated values and this is mainly because of shifts in forecast categories towards the end of the season, which tends to happen if the yield is near a category boundary. For example, 2011 final yield was in the low category

but one month earlier than harvest the ensembles indicate 56 % in the average category and 44 % in the low category (see Figure 11), which results in a low skill score for that year. The overall skill of the system is presented in Figure 19 where it shows a good skill even two months ahead of harvest. The average $RPSS_D$ shows an increase of skill as the lead time decreases, which is expected. Comparison of similar period skill scores for yield forecasted weighted by the IRI seasonal weather forecast of rainfall and temperature showed a similar result to that of the non-weighted forecast. This indicates that the seasonal



forecasts have little impact in predicting the maize yield in the region, which is associated both with the low correlation of seasonal weather values and maize yield and with the vague nature of the forecasts.

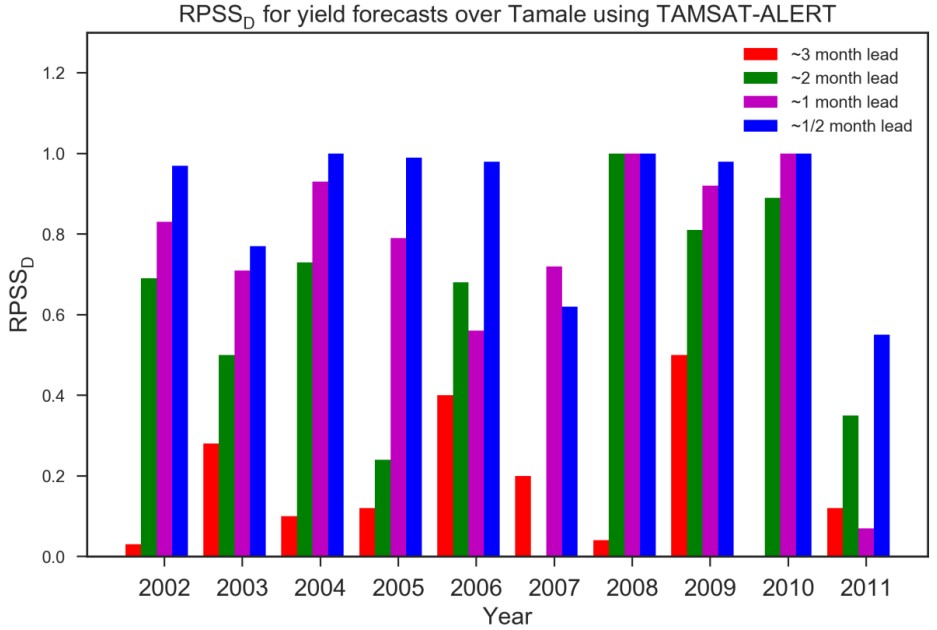

**Figure 18: Discrete ranked probability skill score for the yield forecasts over Tamale using TAMSAT-ALERT system at different lead time.**

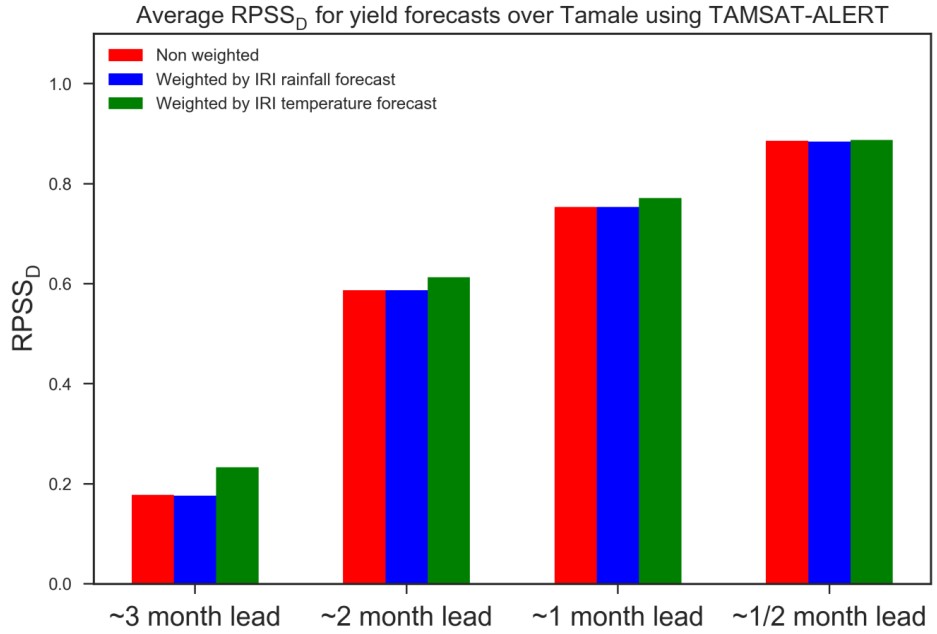

**Figure 19: Discrete ranked probability skill score for the yield forecasts over Tamale using TAMSAT-ALERT system at different lead time averaged for 2002–2011.**





## 4    Discussion and conclusions

The TAMSAT-ALERT framework complements and extends previous systems by driving impact models with ensembles based on observed weather rather than weather generators or direct forcing with seasonal forecasts. This provides a simple means both of combining information at different scales, and of bias correcting seasonal forecasts. The framework is thus
capable of integrating multiple sources of environmental observations/forecasts into continually updated assessments of the likelihood of a user defined adverse event, such as unfavourable weather conditions for maize yield. The system can work at any spatial scale for which driving data is available, including for individual communities.

The use of decision support tools for agricultural activities in Africa is low because of low capacity for model use, lack of funding from governments in the development of agricultural decision support tools, lack of data availability for validation
and calibration of models and low knowledge among decision makers about the use of decision making tools (MacCarthy et al., 2017b). Nevertheless, the demand for meteorologically driven crop models, such as Decision Support System for Agrometeorology Transfer (DSSAT), World Food Studies (WOFOST) and Crop Environment Resource Synthesis–Maize (CERES–Maize) for Sub Saharan Africa, speaks to a need for quantification of the meteorological hazard to yield (Dzotsi et al., 2003, Kassie et al., 2014, Kassie et al., 2015, MacCarthy et al., 2017a). The implementation of TAMSAT-ALERT
described in this study quantifies the meteorological risk to agriculture, and as such potentially provides information for government, aid agencies and nongovernmental organizations working in agriculture.

In the example described in this paper, we have used the GLAM crop model. It is clear from the validation of GLAM against national yield statistics presented in Section 3.2.3, that the model's ability to simulate year to year variation in Ghana-wide yield in maize is moderate. Nevertheless, previous studies have demonstrated that GLAM can capture the meteorological
hazard to yield (Challinor et al., 2007; Challinor et al., 2010; Osborne et al., 2013), when the model is driven with high quality meteorological data and is compared against robust information on yield. The provision of the scripts for the GLAM implementation will enable further studies to be carried out at locations with more robust information on yield and agronomical characteristics.

This study used the GLAM crop model, as an illustration of the implementation of the system. The strength of TAMSAT-
ALERT, however, is its modularity. TAMSAT-ALERT can be implemented for any impact model driven with meteorological data. There is now demand for TAMSAT-ALERT in locations throughout East and West Africa, with the system adapted to implement trusted metrics and models.

A key finding from our study is that tercile probabilistic 90-day rainfall/temperature forecasts have little impact on TAMSAT-ALERT's skill, for the case study considered. This is not unexpected. The correlation of 90-day total rainfall with GLAM
simulated maize yield, in this region, is low. The low correlation means that we do not expect precipitation seasonal forecasts to improve the yield forecasts even if they are skilful. These results are consistent with anecdotal evidence that the tercile regional seasonal forecasts of rainfall routinely issued by regional organizations are of little practical benefit for decision making at local scales. These findings highlight the importance of issuing precisely located forecasts of relevant metrics. Only by considering the links between the meteorological metric being forecast and the hazard being assessed, will the potential
value of skillful sub-seasonal to seasonal forecasts be realized. A secondary application of TAMSAT-ALERT is thus to provide guidance on the design of forecast products that would be of maximal use for decision makers, should they have sufficient skill. Such analyses are currently underway as part of a major national capability programme being carried out at the National Centre for Atmospheric Science.

In summary, TAMSAT-ALERT is a light-weight system, which can be run either using the computing facilities available in
house at meteorological services, or on the cloud. Its modular design enables it to work alongside existing systems to combine





multiple sources of data into quantitative assessments of risk. Together with socio-economic assessments, this information could be of significant value for government, policy makers and humanitarian service providers tasked with mitigating the effect of drought on Africa's poorest farmers.

## 5    Code availability

5    The TAMSAT-ALERT v1.0 frame work code and the user manual are openly available on GitHub (https://github.com/tamsat-alert/v1-0) and (https://doi.org/10.5281/zenodo.1164603). The GLAM-v3 crop model is provided under a license agreement, so it is not possible to directly release it on GitHub but it is possible to obtain it through the contacts form in the following link (http://www.see.leeds.ac.uk/research/icas/research-themes/climate-change-and-impacts/climate-impacts/glam/).

## 6    Acknowledgements

Dagmawi Asfaw is supported by a studentship implemented by CIMMYT as part of Taking Maize Agronomy to Scale in Africa (TAMASA), made possible by the generous support of the Bill and Melinda Gates Foundation (BMGF). Any opinions, findings, conclusion, or recommendations expressed in this publication are those of the author(s) and do not necessarily reflect the view of the BMGF.

15    Emily Black is supported by the NERC/GCRF Official Development Assistance programme, ACREW, which forms part of the core programme of the National Centre for Atmospheric Science - Climate Division. She also gratefully acknowledges support from the BRAVE (NE/M008983/1) and HyCristal (NE/M020371/1) projects. Matthew Brown's and Frederik Otu-Larbi's work on this project was supported by the University of Reading impact programme.  Ewan Pinnington and Tristan Quaife were funded by the UK Natural Environment Research Council project ERADACS (NE/P015352/1) and the National

20    Centre for Earth Observation. Kathryn Jane Nicklin and Andrew Challinor were funded by the NERC/DFID Future Climate for Africa program under the AMMA-2050 project, grant number NE/M020126/1.





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
