# Peer review of "TAMSAT-ALERT v1: A new framework for agricultural decision support"

_Geoscientific Model Development, 2017_

## Referee Comment (RC1) · Anonymous Referee #1 · 14 May 2018

General comments:

This is a highly important and very interesting paper, especially when it comes to strengthening existing early warning systems around the world. Estimating the impacts of weather-related activities (in this case agriculture) will be an invaluable tool for many NGOs and other institutions that aim to mitigate the impact of weather-related disasters. This new framework allows to determine agricultural risk and forecast crop yield outcomes before and during the season. The fact that it can continuously be updated with observed data, and thus reduce uncertainty, is very powerful. And another important result shown here (and not really highlighted by the authors is that the yield can be estimated about 6 weeks prior to harvest.

Specific comments:

[Figure]

- The whole paper is focused on 'low' yield or 'adverse events'. But what about taking advantage of good seasons? There is value in predicting a good season and TAMSAT ALERT can be used in these scenarios.

- The authors set up the system using national data but then somehow only focus on Northern Ghana, while specifying that they do not have yield data for that region. There is mention of Tamale but without clarifying where that is, and what the agricultural practices are in this town/region. This is all rather confusing and it would be good if the authors could somehow clarify this issue.

- Seasonal forecasts do come with uncertainty and it would be good to at least discuss the impact of forecast skill on these results.

- Similarly, agricultural data have some uncertainty, as the authors have indicated in line2 page 5. Discussing the impact on calibrating a crop model using FAO data and thus the impact on the outputs of TAMSAT ALERT should be discussed. Especially, now there is a clear understanding that crop models should not be taken individually (e.g., AgMIP). Additionally, the sensitivity of GLAM to planting/harvest dates could have been included as well. All of this should at least be clearly discussed.

- the authors mention that the forecast is very similar to the climatology, which then obviously results in no clear additional information from the forecast. So, unless I have misunderstood something, I do not believe that the authors can then clearly say that there is limited value in seasonal forecasts. Ideally, and in order to make a clear suggestion on the usefulness of forecasts, the authors should have used several forecasts that use a variety of tercile distributions. So I encourage the authors to be more cautious with their result descriptions.

- the authors focused on 2011 to evaluate this framework. From Fig 5 one can see that GLAM is able to clearly estimate the yield in that year. Is there a way to provide a similar analysis for 2010 where GLAM underperforms? And maybe include more discussion on what the decision-maker needs to do when the year-to-year estimation

of this system is only 'moderate'.

- when including forecast (section 3.3.2), it is unclear whether all forecasts are included simultaneously (i.e., in June, do the authors include forecasts for JJA, JAS, ASO and SON or do they only include JJA and climatology for the rest of the season? Also, is there scope to include both the temperature and rainfall forecasts at the same time?

- And finally, it would have been good to see a quick statistical analysis on the usefulness of the forecasted variables in predicting maize yields in Ghana (i.e. run a multiple linear regression on yield=f(seasonal rainfall, seasonal mean temperature)). This could also provide more arguments while discussing the value of seasonal forecasts.

- As mentioned before, one key result here is that by using climatology alone, in that one year, TAMSAT ALERT can predict maize yields 6-8 weeks ahead of harvest. This is extremely useful for decision makers and a point that should be highlighted more in this paper.

All these points should be included somehow in this paper, they may not all require additional processing and can somehow just be included into the discussion section. And as such, the abstract will then also need to changed.

Technical comments:

page 1 line 13: change to 'which aims to provide early warning'

page 3 line 2: should the question only focus on 'adverse events'? or should it be more general? line 15: again, why focus on 'unfavourable' conditions? line 19: 'to assign that assign' rewrite Figure 1: again focus on 'adverse events'. Overall, figure 1 could be improved so that it clearly highlights all the steps mentioned in 2.2

page 4 line 19: no space between base and line

page 5 line 11: what is the metric period? line 13: remove TAMSAT ALERT line 18: us " for file name line 22: what is ECDF? line 26: use " for file name, and remove comma

after <year>

page 7: line 4: no 'amount of yield' in figure. What is that anyway Figure 3: can you explain the peak in 2002? Figure 4: use same y scale as fig 5

page 9 line 26: no need to write again what WFDEI stands for

page 10 lines 5-6: do you have info on all these other suggested factors?

page 12 (and beyond) you have forgotten to include figure numbers throughout figures 8, 9, 10 all show the same thing, consider using only one Figure 11: not sure what the rationale is to use these dates but I would guess that your figure for Sept 15 would be similar to the one for Oct 4th, and it would indicate the 6-8 week lead time in forecasting yields.

page 16 lines 3-5: this may be simply due to the fact that early season rainfall may have limited impact on yields (something that could technically be evaluated statistically, or at least acknowledged)

page 17 line 13: again, and as mentioned before, it is rather confusing to set the system up for all of Ghana and then only use data for Northern Ghana. Would there be a way to include forecasts for all of Ghana? Fig 17 and 18: they all have different time scales (and fig 4), add why that is.

---

## Referee Comment (RC2) · Anonymous Referee #2 · 16 May 2018

General comments:

This paper highlights an interesting approach to modeling and predicting yield. Overall, the paper is good and has useful information on the new TAMSAT ALERT model and an example of the model in action. The resulting work shows that TAMSAT ALERT can be useful in identifying links between the forecasts and the yield outcomes.

1. In the abstract, the example of Northern Ghana shows that predictions of rainfall and temperature are of limited use to decision makers, but is not followed up in the paper. Specifically, the paper walks through using different data within the TAMSAT ALERT but does not explain why different datasets were used. It would be useful to know if the mean temperature and precipitation forecasts that are issued are incorporated into the model.

[Figure]

2. It seems like this paper might be highlighting an issue of scale- local forecasts may be of more benefit than large, country-wide forecasts. In the conclusion, the benefit of TAMSAT ALERT may also be providing guidance on the design of forecast products (page 24, line 36). Since this may be a secondary use, it might be important to include that at the top of the paper (in both the abstract and the introduction).

3. The target audience of all other early warning platforms are mentioned, but TAMSAT-ALERT's target audience is not mentioned. Perhaps this should be included (page 2 line 22).

Specific comments: 1. The first sentence of the introduction needs a citation (page 1, line 27). My suggestion would be Muller, Cramer, Hare, Lotze-Campen, 2011, "Climate Change Risks for African Agriculture", Proceedings of the National Academy of Sciences. In this paper they talk about naturally high levels of climat variability, reliance on rain-fed agriculture, and limited capacity to cope with climate variability makes Sub-Saharan Africans notably vulnerable.

2. The different platforms available for early warning on the 2nd paragraph of the introduction (pages 1-2) should probably have citations and links for them. Within that comment- the IRI platform is "IRI/LDEO Climate Data Library", and the maprooms are "IRI Climate an Society Maproom"

3. Under the Model Specification, Point 1 (page 4 lines 13-16), the type of data that ALERT can use is not specified. Must it be converted from .csv to .txt? What is the delimiter? Can it accept geotifs or netCDF files?

4. Although TAMSAT-ALERT is designed to be flexible to different inputs, it might be important to include the spatial resolution of TAMSAT precipitation data in this paper, since it seems logical that TAMSAT precipitation data may be one of the most logical inputs?

5. The first time ECDF is mentioned (empirical cumulative distribution function) is on

page 5 line 21, and it is not designated an acronym when it is first mentioned. However, later in the paragraph, (line 22), it is mentioned by acronym. Perhaps the acronym should be designated immediately after the first mention or the acronym on line 22 should be replaced with the name since ECDF is not used again in the paper.

6. On Figure 2 (page 7), there is a pre-existing map of Ghana: this might be a useful figure to include where Tamale is, since this is the example mentioned immediately on page 8 (line 10).

7. On Figues 11e, 12e, 13e, 14e, and 15e - the probability of low yield is at 100%, but the 100% has been cut down to 10. That number should either be scrubbed off or should be 100% and completely visible.

―――――――――――――――――

---

## Author Comment (AC1) · 25 May 2018

Dear Referee,

We thank you very much for your kind and constructive comments.

We have addressed your comments accordingly. The responses and the corrected manuscript are uploaded in the supplement.

Thank you

Please also note the supplement to this comment:
https://www.geosci-model-dev-discuss.net/gmd-2017-316/gmd-2017-316-AC1-supplement.zip

---

## Author Response (AR1)

**Author response to the comments from referee #1**

We thank the author for their kind comments on the importance of this work and for his constructive comments, which we have addressed as described below.

**General comments response:**

| Page number | Line number | Referee comment | Correction made / Sentence added to the paper | Response |
|---|---|---|---|---|
| 25 | 6 | The whole paper is focused on 'low' yield or 'adverse events'. But what about taking advantage of good seasons? There is value in predicting a good season and TAMSAT ALERT can be used in these scenarios. | Page 25 line 6 onwards: Whilst the emphasis on our study has been on forecasting adverse events, such as low yields, it should be noted that TAMSAT-ALERT is also capable of anticipating favourable conditions, enabling decision makers to maximise the benefits of such years - for example by managing post-harvest storage and markets. | Indeed, the system can help to assess events of favorable conditions like high yield and these values may have importance in relation to market price and post-harvest storage issues. But we are more focused on the adverse events because the risk associated with these events are much greater than that of the "good years" for users. A sentence has been added to highlight this point (Page: 25, Line: 6) |
| 9 | 8 | The authors set up the system using national data but then somehow only focus on Northern Ghana, while specifying that they do not have yield data for that region. There is mention of Tamale but without | Page 9, Line 8 onwards We chose to use Tamale because it is in the Northern part of Ghana (Figure 2) where most of the Maize is grown. The station in Tamale also has a long-term record of the driving data for the crop model. It should be noted that TAMSAT-ALERT can in principle be run using any gridded meteorological data like satellite rainfall estimate (E.g. TAMSAT | The available dataset of yield was a country wide average yield for this we used WFDEI dataset to calibrate the crop model used. But the demonstration of the TAMSAT-ALERT system is done using a point gauge data (Tamale) which is indicated on Figure 2, in order to |

| | | | | |
|---|---|---|---|---|
| | | clarifying where that is, and what the agricultural practices are in this town/region. This is all rather confusing, and it would be good if the authors could somehow clarify this issue. | (https://www.tamsat.org.uk/data/rfe/index.cgi) with a resolution of 4 Km (Maidment et al., 2017). | highlight the systems use at the fine scales of relevance for decision makers. A sentence has been added to highlight this point (Page: 9, Line: 8) |
| 19 | 18 | Seasonal forecasts do come with uncertainty and it would be good to at least discuss the impact of forecast skill on these results. | page 19, Line 18 onwards:

In summary, Figures 12 and 13 indicate that if meteorological forecasts have sufficient accuracy and precision, they can add information to the decision-making process, especially in the middle to later part of the growing season.  However, Figures 14-16 show that the tercile forecasts currently issued in northern Ghana do not have sufficient precision to information to yield risk assessments. A further application of TAMSAT-ALERT could be to investigate the level of skill that is required for meteorological forecasts to contribute useful information to such decision-making processes. | We now comment on this at the end of the seasonal forecasting section (page 19, Line 18 onwards) |

| | | | | |
|---|---|---|---|---|
| 25 | 29 | Similarly, agricultural data have some uncertainty, as the authors have indicated in line2 page 5. Discussing the impact on calibrating a crop model using FAO data and thus the impact on the outputs of TAMSAT ALERT should be discussed. Especially, now there is a clear understanding that crop models should not be taken individually (e.g., AgMIP). Additionally, the sensitivity of GLAM to planting/harvest dates could have been included as well. All of this should at least be clearly discussed. | Page 25, Line 29 onwards
This modularity and flexibility is important, since the skill of the TAMSAT-ALERT system is constrained by the quality of the model and its calibration. In this study, for example, the evaluation and calibration of GLAM was hampered by quality-control issues with the available yield data. The system would be much improved if used in-house by agencies with access to high quality yield data, and locally calibrated models. | Indeed, the skill of seasonal forecast and the skill of the model used has an impact in the outcome of the risk assessment. Hence, using a well calibrated model is important and a sentence has been added to highlight this point (Page: 25, Line: 29 onwards). |
| 26 | 1 | the authors mention that the forecast is very similar to the climatology, which then obviously results in no clear additional information from the forecast. So, unless I have misunderstood something, I do not believe that the authors can then clearly say that there is limited value in seasonal forecasts. | Page: 26, Line: 1 onwards
Our results do not suggest that there is no information available from seasonal forecasts. However, we do show that 90-day tercile forecasts of temperature and rainfall, even if perfectly skillful, provide comparatively little information to risk assessments for low maize yield. This could be because the sensitivity of crops to moisture is on a specific period of their growth and the sensitivity of crops to temperature is also not similar throughout their growth stage. In other words, our findings highlight the | The results shown in Figures 12 and 13 show that even if a perfectly precise tercile forecast is used, the information provided to risk assessments is minimal. This is because the link between the metric being forecast (90 day averages of rainfall/temperature) are not strongly correlated with yield. We are not claiming that seasonal forecasts cannot add information to risk assessment, but |

| | | | | |
|---|---|---|---|---|
| 19 | 18 | Ideally, and in order to make a clear suggestion on the usefulness of forecasts, the authors should have used several forecasts that use a variety of tercile distributions. So I encourage the authors to be more cautious with their result

descriptions. | necessity of more specific and localized forecasts, if users are to benefit from the inherent skill contained in the forecasts.

Page 19 line 18 onwards
In summary, Figures 12 and 13 indicate that if meteorological forecasts have sufficient accuracy and precision, they can add information to the decision-making process, especially in the middle to later part of the growing season.  However, Figures 14-16 show that the tercile forecasts currently issued in northern Ghana do not have sufficient precision to information to yield risk assessments. A further application of TAMSAT-ALERT could be to investigate the level of skill that is required for meteorological forecasts to contribute useful information to such decision-making processes. | rather we are highlighting the need for seasonal forecast to tailored to their application. Sentence have been added to clarify this point (Page: 26, Line: 1) and (Page: 19, Line: 18). |
| 25 | 29 | the authors focused on 2011 to evaluate this framework. From Fig 5 one can see that GLAM is able to clearly estimate the yield in that year. Is there a way to provide a similar analysis for 2010 where GLAM underperforms? And maybe include more discussion on what the

decision-maker needs to

do when the year-to-year | page 25 line 29 onwards.

This modularity and flexibility is important, since the skill of the TAMSAT-ALERT system is constrained by the quality of the model and its calibration.  In this study, for example, the evaluation and calibration of GLAM was hampered by quality-control issues with the available yield data. The system would be much improved if used in-house by agencies with access to high quality yield data, and locally calibrated models. Nevertheless, it is important that model error is taken into account in the decision-making process, and forecasts should therefore be issued in the | Indeed, if it is to provide useful information, TAMSAT-ALERT must incorporate models that are capable of simulating accurately the metric under assessment. This is clarified in the discussion section, along with brief discussion of strategies that users might adopt to deal with model error. |

| | | estimation of this system is only 'moderate'. | context of model evaluations like the one presented in this study. TAMSAT-ALERT's modular structure, moreover, permits forecasts to be produced using an ensemble of crop models/crop model parameterizations - facilitating formal analysis of model uncertainties. | |
|---|---|---|---|---|
| 14 | 18 | A. when including forecast (section 3.3.2), it is unclear whether all forecasts are included simultaneously (i.e., in June, do the authors include forecasts for JJA, JAS, ASO and SON or do they only include JJA and climatology for the rest of the season?

B. Also, is there scope to include both the temperature and rainfall forecasts at the same time? | The forecasts are commonly issued at the start of every month. Hence, we have applied the forecasts only to the meteorological season being forecasted with the remaining season not included in the weighting estimation. For example, for running TAMSAT-ALERT on June 4, the seasonal forecast of June-July-August is applied. | A. A sentence has been added to clarify (Page: 14, Line: 18).

B. Thank you - this is a good idea! In principle it is possible to include multiple forecasts in the system which we will try to include in the next version. For now, users can only use a single weighting variable of their choice (not only rainfall or temperature but other forecasts variables available to them like nino3.4). However, as we have showed in the results the weighting variables need to have a very strong correlation with the metric being evaluated for the |

| | | | | weighting to have an impact on the decision made. |
|---|---|---|---|---|
| 19 | 14 | And finally, it would have been good to see a quick statistical analysis on the usefulness of the forecasted variables in predicting maize yields in Ghana (i.e. run a multiple linear regression on yield=f(seasonal rainfall, seasonal mean temperature)). This could also provide more arguments while discussing the value of seasonal forecasts. | ...the relationship between the seasonal cumulative rainfall and seasonal mean temperature with maize yield is very low (see supplementary document Figure S1 and Figure S2). | The figures indicating the correlation between seasonal rainfall and yield as well as seasonal mean temperature with yield have been added to the supplementary document. A sentence has been added referring to this in the paper (Page: 19, Line 14) |
| 25 | 19 | As mentioned before, one key result here is that by using climatology alone, in that one year, TAMSAT ALERT can predict maize yields 6-8 weeks ahead of harvest. This is extremely useful for decision makers and a point that should be highlighted more in this paper. | Page: 25, Line: 19

A key result is that, even in the absence of meteorological seasonal forecasts, low yield can be anticipated 6-8 weeks before with some skill. | Thank you for mentioning this. A sentence has been added to point out this result (Page: 25, Line: 19). |

**Technical comments response:**

| Page number | Line number | Referee comment | Correction made / Sentence added to the paper | Response |
|---|---|---|---|---|
| 1 | 13 | page 1 line 13: change to 'which aims to provide early warning' | Which aims to provide early warning | Corrected! |
| 3 | 2 | A. page 3 line 2: should the question only focus on 'adverse events'? or should it be more general? line 15: again, why focus on 'unfavourable' conditions? | | A. See above in the general comments! |
| 3 | 19 | B. line 19: 'to assign that assign' | | B. Corrected! |
| 4 | 5 | C. rewrite Figure 1: again focus on 'adverse events'. Overall, figure 1 could be improved so that it clearly highlights all the steps mentioned in 2.2 | | C. We do not want to congests the framework flow; that is why we specify the inputs processes and outputs in the chart (Figure 1) and provide more detailed explanation in section 2,2. In addition, a detailed User manual has been available with the code (https://github.com/tamsat-alert/v1-0) as part of the requirement of the Journal. But we have edited the figure with different colors to highlight inputs, processes and output. |

| | | | | |
|---|---|---|---|---|
| 5 | 15 | A. page 5 line 11: what is the metric period? | A. The metric period is the period on which the weighting will be done, and the probabilistic risk is calculated. For example, if one wants to estimate the metrological risk on available soil moisture the ensembles can be run for a lot longer period to allow spin up of model to equilibrium values for initial condition required but, the main interest for the user might be the first 90 days hence the length of the metric period is only the first 90 days and all the risk analysis is done on this metric period. | A. A definition has been added (Page: 5, Line: 15) onwards. |
| | | B. line 13: remove TAMSAT ALERT line 18: use" for file name line 22: | | B. Corrected! |
| 5 | 30 | C. what is ECDF? line | | C. Full description of the acronym has been mentioned (Page: 5, Line: 30). |
| | | D. 26: use " for file name, and remove comma after <year> | | D. Corrected! |
| | | | B. The empirical cumulative distribution function (ECDF) | |
| 3 | 19 | A. page 7: line 4: no 'amount of yield' in figure. What is that anyway Figure 3: can you explain the peak in 2002? | | A. The figure shows only the total production area of maize in Ghana. It was only presented to show that that the production area is increasing to emphasize the importance of maize crop. We do not have any information on the |

| | | | | increase in production area in the 2002 season. |
|---|---|---|---|---|
| | | B. Figure 4: use same y scale as fig 5 | | |
| | | | | B. Corrected! |
| 9 | 26 | page 9 line 26: no need to write again what WFDEI stands for | | Corrected! |
| 10 | 5 | page 10 lines 5-6: do you have info on all these other suggested factors? | | We do not have any detailed information, but TAMSAT-ALERT is only explaining the meteorological risk to yield. But yield can be impacted by the factors suggested hence, the line was added to make users cautious when interpreting the results form TAMSAT-ALERT. |
| 9 | 8 | Where is Tamale and why only northern Ghana? | Page 9 line 8

We chose to use Tamale because it is in the Northern part of Ghana (Figure 2) where most of the Maize is grown. The station in Tamale also has a long-term record of the driving data for the crop model. | This is clarified in the text. |
| 12 | | A. page 12 (and beyond) you have forgotten to include figure numbers throughout figures 8, 9, 10 all show the same thing, consider using only one | | A. This error was on the first manuscript submitted the one available online do not have this error. (this was corrected on the first topical editor comments were **corrected**!). |

| | | | | |
|---|---|---|---|---|
| | | B. Figure 11: not sure what the rationale is to use these dates but I would guess that your figure for Sept 15 would be similar to the one for Oct 4th, and it would indicate the 6-8 week lead time in forecasting yields. | | B. There is no limit on the day range to run the TAMSAT-ALERT system. The dates are chosen to show yield changes every 4 weeks starting from planting to harvest. But the general conclusion is we can say something about what the yield is going to be 6 – 8 weeks prior to harvest. A sentence has been added to highlight this see general comment response. |
| 16 | 10 | page 16 lines 3-5: this may be simply due to the fact that early season rainfall may have limited impact on yields (something that could technically be evaluated statistically, or at least acknowledged) | The improvement is less noticeable in June and July, perhaps reflecting the fact that, at least in the GLAM crop model, cumulative rainfall in this part of the season is comparatively less strongly correlated with yield. | A sentence has been added to acknowledge. (Page: 16, Line: 10). Relationship of seasonal rainfall with yield is also shown in the supplementary document (see supplementary document Figure S1 and Figure S2). (See general comment response above). |
| | | A. page 17 line 13: again, and as mentioned before, it is rather confusing to set the system up for all of Ghana and then only use data for Northern Ghana. Would there be a way to include forecasts for all of Ghana? | A. ---

 B. Page 22 Lines 14 onwards
 Only 2007-2012 are presented in Figure 17 because the maize variety changed in 2007, making the hindcasts of these years more relevant to the present day | A. Please see the response in the general comment above!

 B. A sentence has been added to the paper why we only show the last five years (Page: 22, Line: 14).
 C. Figure 4 shows the Average yield over Ghana, but only Statistical |

| 22 | 14 | B. Fig 17 and 18: they all have different time scales? (and fig 4), add why that is. | than the 1994-2006 period (see supplementary document Figure S3) | analysis was done from 2002 only (Figure 18) because some of the Gauge data in 1998 and 1999 has some missing values so we do not want to include the results in the skill analysis even though we used climatology values to run the system. |

**Author response to the comments from referee #2**

We thank the reviewer for their recognition of the value of this work and for their constructive comments, which we have addressed as described below.

**General comments response:**

| Page number | Line number | Referee comment | Correction made / Sentence added to the paper | Response |
|---|---|---|---|---|
| 19 | 18 | 1. In the abstract, the example of Northern Ghana shows that predictions of rainfall and temperature are of limited use to decision makers, but is not followed up in the paper. Specifically, the paper walks through using different data within the TAMSAT ALERT but does not explain why different datasets were used. It would be useful to know if the mean temperature and precipitation forecasts that are issued are incorporated into the model. | Page 19 Line 18 onwards:

In summary, Figures 12 and 13 indicate that if meteorological forecasts have sufficient accuracy and precision, they can add information to the decision-making process, especially in the middle to later part of the growing season. However, Figures 14-16 show that the tercile forecasts currently issued in northern Ghana do not have sufficient precision to information to yield risk assessments. A further application of TAMSAT-ALERT could be to investigate the level of skill that is required for meteorological forecasts to contribute useful information to such decision-making processes. | We used the WFDEI data set to evaluate the GLAM crop model over Ghana since the yield data we got is from all Ghana maize average from FAO. So, we evaluate and calibrate the model, but the evaluation of TAMSAT-ALERT system was evaluated using gauge data from Tamale. The reason is not to introduce more error due to the satellite data estimates of driving forces. But, we will evaluate the system using different data set in the future work.

We have also extended the discussion of the use of seasonal forecasts within TAMSAT-ALERT. |
| 1 | 27 | 2. It seems like this paper might be highlighting an issue of scale-local forecasts may be of more benefit than large, country-wide forecasts. In the conclusion, the benefit | This finding speaks to the pressing need for meteorological forecast products that are tailored for individual user applications. | Sentences were added in the abstract (Page 1, Line 27). The issue of providing bespoke forecast metrics is also addressed in greater |

| 26 | 1 | of TAMSAT ALERT may also be providing guidance on the design of forecast products (page 24, line 36). Since this may be a secondary use, it might be important to include that at the top of the paper (in both the abstract and the introduction). | Page 26 line 1 onwards: Our results do not suggest that there is no information available from seasonal forecasts. However, we do show that 90-day tercile forecasts of temperature and rainfall, even if perfectly skillful, provide comparatively little information to risk assessments for low maize yield. This could be because the sensitivity of crops to moisture is on a specific period of their growth and the sensitivity of crops to temperature is also not similar throughout their growth stage. In other words, our findings highlight the necessity of more specific and localized forecasts to benefit from inherent skills contained in the forecasts. | detail in the discussion (Page 26 Line 1) to highlight the point. |
| 2 | 32 | 3. The target audience of all other early warning platforms are mentioned, but TAMSATALERT's target audience is not mentioned. Perhaps this should be included (page 2 line 22). | The impact model output and the weather risk associated with the output that can be obtained from TAMSAT-ALERT can be used by government, non-governmental organizations involved with providing farming information and aid, and weather index insurance providers can be benefited from continuous assessment of the risk. | Sentence was added to state target audience of the system (Page 2 Line 32). |

**Technical comments response:**

| Page number | Line number | Referee comment | Correction made / Sentence added to the paper | Response |
|---|---|---|---|---|

| 1 | 31 | 1. The first sentence of the introduction needs a citation (page 1, line 27). My suggestion would be Muller, Cramer, Hare, Lotze-Campen, 2011, "Climate Change Risks for African Agriculture", Proceedings of the National Academy of Sciences. In this paper they talk about naturally high levels of climat variability, reliance on rain-fed agriculture, and limited capacity to cope with climate variability makes Sub-Saharan Africans notably vulnerable. | Many African people depend on rain–fed agriculture, and are thus vulnerable to drought, and other weather–related hazards exacerbated by climate change (Muller et al., 2011). | Corrected! |
| --- | --- | --- | --- | --- |
| 2 | | 2. The different platforms available for early warning on the 2nd paragraph of the introduction (pages 1-2) should probably have citations and links for them. Within that comment- the IRI platform is "IRI/LDEO Climate Data Library", and the maprooms are "IRI Climate an Society Maproom" | The Rainwatch-AfClix early warning system (RWX) (http://www.rainwatch-africa.org/rainwatch/), Famine Early Warning Systems Network Early Warning Explorer (FEWSNET-EWX) (https://earlywarning.usgs.gov/fews/ewx/index.html) International Research Institute (IRI) data library/map rooms (http://iridl.ldeo.columbia.edu/index.html?Set-Language=en), Africa Flood and Drought Monitor (AFDM) (http://stream.princeton.edu/AWCM/WEBPAGE/interface.php), | Corrected! |
| 4 | 13 -16 | 3. Under the Model Specification, Point 1 (page 4 lines 13-16), the type of data that | | The TAMSAT-ALERT system we have takes data from text files and reproduce it in the |

| | | | | |
|---|---|---|---|---|
| | | ALERT can use is not specified. Must it be converted from .csv to .txt? What is the delimiter? Can it accept geotifs or netCDF files? | | format required by the crop model used GLAM. When using a different model the the data should be given in the format required by the model incorporated in the system. A user Manual is provided on how to run TAMSAT-ALERT and what data format you can use. See Code Availability: ([https://github.com/tamsat-alert/v1-0](https://github.com/tamsat-alert/v1-0)). |
| 9 | 8 | 4. Although TAMSAT-ALERT is designed to be flexible to different inputs, it might be important to include the spatial resolution of TAMSAT precipitation data in this paper, since it seems logical that TAMSAT precipitation data may be one of the most logical inputs? | Page 9, line 8 onwards: We chose to use Tamale because it is in the Northern part of Ghana (Figure 2) where most of the Maize is grown. The station in Tamale also has a long-term record of the driving data for the crop model. | Sentence has been added explaining the possibilities of using gridded data set and the resolution of the TAMSAT rainfall data (Page 9  Line  8). |
| 5 | 30 | 5. The first time ECDF is mentioned (empirical cumulative distribution function) is on page 5 line 21, and it is not designated an acronym when it is first mentioned. However, later in the paragraph, (line 22), it is mentioned by acronym. Perhaps the acronym should be designated immediately after the first mention or the acronym on line 22 | The empirical cumulative distribution function (ECDF) | Corrected! |

| | | | | |
|---|---|---|---|---|
| | | should be replaced with the name since ECDF is not used again in the paper. | | |
| 7 | | 6. On Figure 2 (page 7), there is a pre-existing map of Ghana: this might be a useful figure to include where Tamale is, since this is the example mentioned immediately on page 8 (line 10). | | Corrected! |
| | | 7. On Figures 11e, 12e, 13e, 14e, and 15e - the probability of low yield is at 100%, but the 100% has been cut down to 10. That number should either be scrubbed off or should be 100% and completely visible. | | Corrected! |

[revised manuscript text omitted]